# Optimizing Landscape Structure of Hybrid Land Use in Ecological Corridors Based on Comprehensive Benefit Index in Metropolitan Area

**Jiake Shen [1] and Yuncai Wang [1,2,*]**

1 College of Architecture and Urban Planning, Tongji University, Shanghai 200092, China
2 Center of Ecological Planning and Environment Effects Research, Joint Laboratory of Ecological Urban Design, Shanghai 200092, China
* Correspondence: wyc1967@tongji.edu.cn; Tel.: +86-021-65980253

**Abstract:** As an inherent part of the landscape ecological network (LEN), the ecological corridor is the bridge between ecological sources and also the key to ensuring urban ecological security. Existing studies on ecological corridors mostly stay in the large scale of landscape patterns and lack guidance for corridor structure design or optimization at the land use level. To provide a reference for the internal landscape structure adjustment of the ecological corridor composed of hybrid land use in the rapidly urbanized areas, first, we constructed the "Comprehensive benefit index of ecological corridors I" by using the three indexes of "Ecological benefit enhancement potential", "Economic input cost" and "Social coordination cost". Second, with the goal of maximizing the comprehensive benefits of the three aspects of ecological corridor construction, we established a functional relationship between the converted agricultural land area $A$, constructed land area $C$, and index $I$ to determine the optimal proportion of agricultural lands and constructed lands converted into ecological lands within the planning scope of the ecological corridors. The results show that (1) according to the conversion ratio, the ecological corridors in the study area can be divided into three degrees of conversion rate: low, moderate, and high. (2) Among the 66 ecological corridors, the agricultural lands in 26 ecological corridors and the constructed lands in 35 ecological corridors need to be converted into ecological land at a high ratio to ensure the comprehensive benefits of the corresponding corridors. We also put forward suggestions for landscape structure adjustment and optimization for ecological corridors with different conversion degrees. This method can help balance the benefits of different stakeholders in the city and implement the results of ecological corridor planning on a large spatial and temporal scale at the land use level.

**Keywords:** ecological corridor structural optimization; mixed landscape land; comprehensive benefit index; land use optimal conversion ratio; metropolitan area



## 1. Introduction

Ecological land is the land with important ecological functions and provides ecosystem services essential for the maintenance of regional ecological security [1], which not only supports and protects the ecosystem stability, the virtuous cycle, and sustainable development [2] but also provides the space necessary for basic ecological activities for the inhabitants of metropolitan areas [3]. Although there is currently no uniform definition of ecological land, it has been identified as a distinct land type in many studies [4–6], especially in metropolitan areas. However, many metropolitan areas today face ecological risks resulting from the degradation and fragmentation of ecological lands, as well as the high dependence and demand of urban residents for ecosystem services provided by these ecological lands [7]. The ecological environment is a complex of natural factors that has an impact on human survival and development [8]. As an important component and

constitution, the degradation and loss of ecological lands seriously threaten the stability of the ecological environment and the well-being of the human beings living in it.

In this context, the study pointed out that constructing a landscape ecological network has emerged as an effective approach to improving the ecological environment and achieving regional sustainable development [9]. Currently, a significant number of landscape ecological networks have been implemented as urban ecological land planning strategies at large temporal and spatial scales, both in research and practice [10–12]. The construction of landscape ecological networks is primarily based on the traditional approach of identifying existing ecological spaces as ecological source areas and establishing ecological corridors to connect these source areas [13]. Ecological corridors are strip-like or linear ecological landscapes with a certain spatial range [14,15], which are heterogeneous compared with the surrounding landscape [16]. They act as bridges and connections between various ecological source areas and are also crucial for achieving urban sustainable development [17]. Unlike the linear transportation infrastructure and ecological roads in the city, which also play a role in connecting destinations and delivering goods and materials, ecological corridors also play the functions of natural habitat, green open space, human habitat isolation, and human settlement environment beautification and enhancement [16,18], which are an inherent part of the landscape ecological network [16] and one of the concrete embodiments of the ecological environment in the metropolitan areas. Consequently, how to reasonably construct ecological corridors has become an important approach to maintaining connectivity between isolated habitat patches and mitigating habitat fragmentation in metropolitan areas [19].

Existing research on ecological corridors primarily focuses on pathway planning and identification. Conventional methods such as minimum cumulative resistance (MCR) analysis [20,21], circuit theory [22–24], graph theory [25,26], morphological spatial pattern analysis (MSPA) [21,27], and suitability/sensitivity analysis [28,29] are employed to identify suitable geographic locations and specific pathways for ecological corridors between different ecological patches. This research remains at the large-scale landscape pattern level, with an emphasis on establishing structural connectivity within landscape ecological networks between isolated ecological patches [30]. The ecological corridors determined using the aforementioned methods are grid pathways with a width of 1, lacking true/actual width and scope [26], which results in limited effectiveness in guiding the scale of ecological corridor construction [16]. As a result, this research fails to translate the construction of ecological corridors in metropolitan areas into specific landscape land at a smaller scale, thus limiting its ability to guide the engineering implementation of ecological corridors.

Another aspect of research on ecological corridors focuses on identifying their planning scope with the objective of biodiversity conservation or the protection of specific species [31–33]. Additionally, some studies develop suitable construction widths for ecological corridors based on new models that consider the comprehensive impact of specific urban environmental factors, whether promoting or limiting [34]. These studies establish clear spatial boundaries for ecological corridors, advancing the implementation of their construction further. Given that ecological corridors are primarily situated in crucial areas of urban ecosystems, their planning pathways and scope inherently incorporate a substantial portion of existing land uses and hybrid landscapes within metropolitan areas [34]. However, the aforementioned research has not yet delved into in-depth discussions regarding corridor design strategies concerning specific land uses within the planning scope.

In highly urbanized areas, ecological lands are widely occupied, and it is difficult to maintain normal ecological processes only through the protection of existing ecological lands [35]. Therefore, it is sometimes necessary to adjust parts of the existing development lands or convert these non-ecological lands back into ecological lands to achieve the goal of ecological corridor construction [36]. Ideally, converting all non-ecological lands within the corridor boundaries into ecological lands would undoubtedly maximize the overall ecological benefits of ecological corridors and landscape ecological networks. However, in reality, one must consider the economic and social costs associated with land conversion.

In metropolitan areas, the construction of ecological corridors involves various interest groups [37]. For example, the process of creating new ecological lands inevitably involves resource and financial investments, as well as the coordination or even relocation of existing residents [38]. Therefore, in the face of the contradiction between economic development and ecological conservation in the context of urbanization, as well as the emergence of urban environmental issues and the growing human ecological needs, a spatial pathway that balances ecological conservation and economic development must be adopted in the design of ecological corridors [16] to achieve the maximum benefit with minimum investment [16,34].

Some studies have carried out corridor construction through the adjustment of land use within ecological corridors. On the basis of reconstructing the ecological security pattern of the Su-Xi-Chang metropolitan area, some scholars have proposed specific plans to reduce the construction of land in the ecological corridor [39]. Other scholars have identified the total area where non-ecological land needs to be converted into ecological land in the main urban area of Chongqing [40]. Li, et al. [41] proposed that a 200 m-wide corridor in Nanjing is ecologically suitable and economically sound for land use adjustment. However, there is still a gap in understanding how to effectively incorporate existing land uses into the adjustment process within the designated width of ecological corridors to achieve a balanced and maximized comprehensive benefit in the context of multidimensional and complex ecological, economic, and social environments. This requires establishing effective mechanisms to evaluate the comprehensive impact of ecological, economic, and social benefits on the construction of urban ecological corridors through land use adjustment.

Addressing the limitations and gaps in previous research on ecological corridors in metropolitan areas, as well as the practical considerations involved in constructing them within rapidly urbanizing regions, this study introduces a new indicator called the "Comprehensive Ecological Corridor Benefit Index". This index measures/quantifies the comprehensive benefits in terms of ecological, social, and economic aspects achieved through adjusting existing land uses and constructing ecological corridors in urban built environments. By maximizing the Comprehensive Ecological Corridor Benefit Index, we have identified the optimal conversion ratio for agricultural land and constructed land within the ecological corridor planning scope to be converted into ecological land. This finding serves to guide the optimization of hybrid land use and landscape structure within the planning scope of ecological corridors in metropolitan areas. This index and methodology provide quantitative references and guidance for scientists, environmental practitioners, and land planners in optimizing the structural composition of existing hybrid landscape land within ecological corridors [42]. Moreover, it facilitates the implementation of ecological corridor planning outcomes at large spatial and temporal scales into specific land uses and supports the construction and development phases.

## 2. Materials and Methods

### 2.1. Methodological Framework

In metropolitan areas, the planning scope of ecological corridors typically encompasses hybrid landscapes composed of existing ecological, agricultural, and constructed lands (Figure 1—green box). Different stakeholders, such as residents, farmers, government agencies, and environmental conservationists, with their respective interests, activities, and decisions, collectively influence the comprehensive benefits of ecological corridors, which are reflected in the ecological, economic, and social dimensions [16,38]. In other words, by adjusting the ratios of different types of landscape land within the planning scope of ecological corridors, it is possible to regulate the benefits achieved in the three dimensions through the construction of ecological corridors.

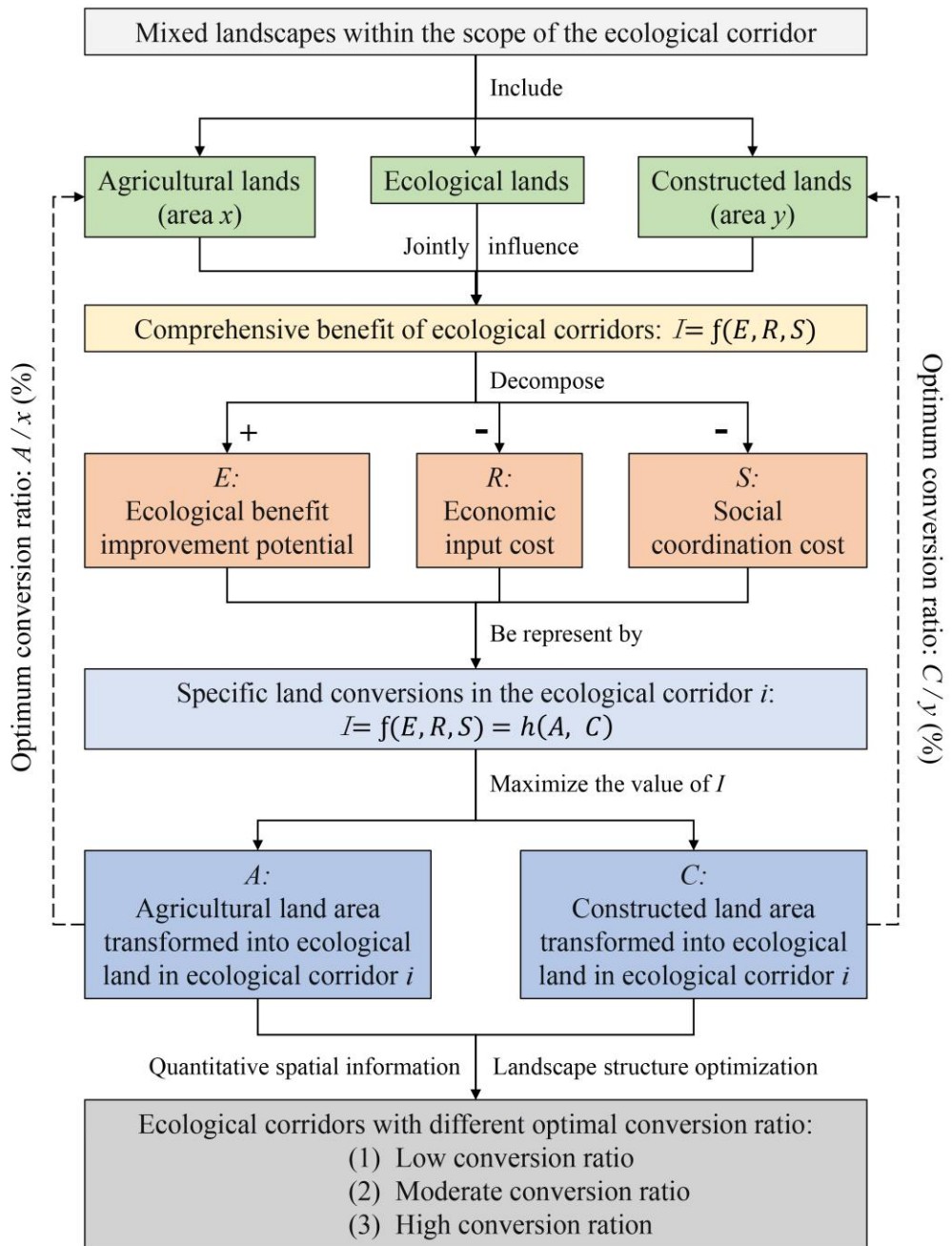

**Figure 1.** Methodological framework of the study.

Based on the aforementioned mechanism, we have developed a new indicator in this study, referred to as the "Comprehensive Ecological Corridor Benefit Index (*I*)" (Figure 1—yellow box), which represents the comprehensive performance of the three aspects of benefits achieved through the construction of ecological corridors. This indicator is a function of three factors (Figure 1—orange box): (1) "Ecological benefit enhancement potential (*E*)", which represents the ecological benefits achieved through ecological corridor construction. When existing cultivated land or constructed land is converted to ecological land, its capacity to provide multiple ecosystem services will be enhanced, bringing additional ecological benefits to the ecological corridors. (2) "Economic input cost (*R*)", which represents the economic costs associated with the construction of the ecological corridor. The conversion between different types of land use is a process of mutual conversion or competition for control and coverage of space, which is achieved by overcoming various resistances [10]. In the process of converting non-ecological land to ecological land, it is

necessary to overcome the "ecological resistance" caused by ecological factors such as topography, slope, vegetation cover, and land cover [43,44]. The process of overcoming these ecological resistances means that construction costs are commensurate. (3) "Social coordination cost (*S*)", which represents the social costs involved in the construction of the ecological corridor. When non-ecological land within an ecological corridor is converted to ecological land, indigenous peoples and stakeholders are implicated [37], requiring the government to compensate or evict it at a cost [38]. Therefore, the total population in non-ecological lands within ecological corridors that need to be converted into ecological lands can reflect the social coordination costs in the process of building ecological corridors through land use adjustment. It should be noted that *E* is directly proportional to *I*, while *R* and *S* are inversely proportional to *I*. The purpose of constructing this functional relation is to find *E*, *R*, and *S* when *I* is maximized.

To further clarify the suitable area and ratio of the conversion from non-ecological lands to ecological lands within the ecological corridor and to provide spatial quantification information for the land structure design within the planning scope of the ecological corridor, we utilized the agricultural land area (*A*) and constructed land area (*C*) as secondary indicators to represent the three primary indicators, namely, *E*, *R*, and *S* (Figure 1—light blue box). We calculated *E* by multiplying the average improvement of multiple ecosystem services per unit area of cultivated land (constructed land) per unit area in the ecological corridor and *A* (*C*). *R* was calculated by the product of the modified ecological resistance value of cultivated land (constructed land) per unit area in the ecological corridor and *A* (*C*). *S* was calculated by the product of population density of the study area and *A* (*C*). This step established the functional relationship between the agricultural land area (*A*) and constructed land area (*C*) converted into ecological lands within the ecological corridor and the "Comprehensive Ecological Corridor Benefit Index (*I*)". This allows us to further determine the optimal agricultural and constructed land areas and their ratio within the corridor that maximizes the overall Comprehensive Ecological Corridor Benefit Index (*I*) (Figure 1—dark blue box). This area ratio can provide a reference for the quantifiable threshold for the conversion of non-ecological land to ecological land within the ecological corridor, thereby simplifying the above complex question of how much cultivated land/constructed land needs to be converted into ecological land in an ecological corridor to maximize the comprehensive benefits of the corridor?

In the end, based on the optimal conversion ratios of agricultural land and constructed land within the planning scope of ecological corridors, all ecological corridors are classified into three types: low, medium, and high conversion degree ecological corridors (Figure 1—dark gray box). Each type of ecological corridor with different conversion degrees corresponds to specific landscape structural optimization strategies and land use adjustment measures. This will provide tailored structural design strategies and optimization guidance for ecological corridors located in different locations, with varying widths, and composed of different hybrid landscapes. For the methodological framework of this study, please see Figure 1.

### 2.2. Study Area

The Su-Jia-Hu area is a collective term referring to the cities of Suzhou, Jiaxing, and Huzhou. It is part of the Greater Taihu Lake Urban Agglomeration, with each city located to the east or south of Taihu Lake. Suzhou is situated in Jiangsu Province, while Jiaxing and Huzhou are located in Zhejiang Province (please see Figure 2). The Su-Jia-Hu area has a total area of 18,221 km$^2$ and a permanent population of 18,315 million. The region enjoys convenient transportation located at a prime geographical position and is situated in the heart of the Yangtze River Delta urban agglomeration. It is in close proximity to major metropolises such as Shanghai, Nanjing, and Hangzhou and, therefore, benefits from the influence radiated by Shanghai. The Su-Jia-Hu area is located in the Taihu Lake Basin and is characterized by a network of rivers and abundant aquatic ecosystems. The overall river density in the region exceeds 1.5 km/km$^2$. As an integral part of the Taihu

Lake Basin, the water bodies and other ecological lands within the Su-Jia-Hu area provide essential ecosystem services for urban areas. Since the initiation of economic reforms and opening-up, the Su-Jia-Hu area has undergone rapid urbanization, accompanied by substantial economic growth and population expansion. Today, it has emerged as one of the most developed areas in China, characterized by robust investment growth and vibrant social development [45].

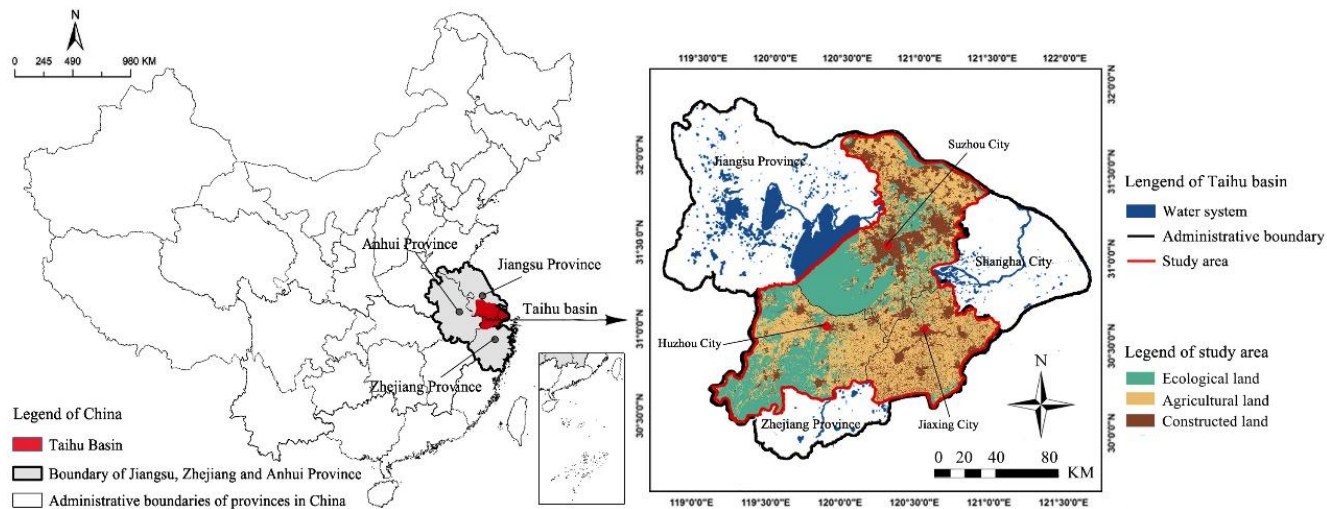

**Figure 2.** The location of the study area and its three types of land use distribution map.

The abundant ecological environment, coupled with rapid economic development and population growth, has brought about significant and pressing challenges in the Su-Jia-Hu area concerning the intricate interplay between human society and the ecological environment. This has resulted in various ecological risks and environmental issues, such as land resource encroachment, agricultural and industrial pollution, and fragmentation of ecological lands, which are typical manifestations of urbanization. These issues pose a serious threat to the ecological security of the Su-Jia-Hu area. Therefore, in this region, achieving balanced, comprehensive benefits in terms of ecology, society, and economy is the primary objective. It is crucial to undertake structural design and land use adjustment for the planned ecological corridors based on the existing local land use and landscape spatial patterns to accomplish this. This is of utmost importance for constructing landscape ecological networks and implementing ecological development in the area.

*2.3. Data Sources and Processing*

In this study, a total of five databases were utilized to calculate the optimal degree of conversion from non-ecological to ecological lands within the ecological corridors in the study area. Among them, two databases were derived from previous studies conducted by the research team in the same study area, including spatial data on 66 ecological corridors and their widths [34], as well as spatial data on the demand levels for three ecosystem services (flood regulation, local climate regulation, outdoor recreation) in the constructed lands of the study area [46]. Other datasets used in the study include land use data (2015), China's population in grid transformation (2015), and a spatial distribution dataset of ecosystem service values in terrestrial ecosystems in China (2015). The sources, types, and applications of the data in this study are summarized in Table 1.

**Table 1.** Five databases, sources, types, and applications used in the study.

| Data | Source | Type | Usage |
|---|---|---|---|
| 66 ecological corridors and their widths | [34] | Shapefile Polygon | Used to provide the planning scope of each ecological corridor |
| Spatial data on the demand levels for three ecosystem services (flood regulation, local climate regulation, and outdoor recreation) in the constructed lands of the study area | [46] | GRID 15 m × 15 m | Used to calculate the economic input cost $R$: to correct the value of ecological resistance of constructed lands |
| Land use data (2015) | Beijing Digital View Technology Co., Ltd., Beijing, China (http://www.dview.com.cn/) (accessed on 20 July 2023) Geographical Information Monitoring Cloud Platform (http://www.dsac.cn/) (accessed on 20 July 2023) | GRID 15 m × 15 m | Used to calculate the agricultural land area $x$ and constructed land area $y$ within the planning scope of ecological corridor |
| China's population in grid transformation (2015) | Geographic Data Sharing Infrastructure, College of Urban and Environmental Science, Peking University (http://geodata.pku.edu.cn) (accessed on 20 July 2023) | GRID 1 km × 1 km | Used to calculate the social coordination cost $S$: total population and population density |
| Spatial distribution dataset of ecosystem service values in terrestrial ecosystems in China (2015) | Resource and Environment Science and Data Center, https://www.resdc.cn/data.aspx?DATAID=258 (accessed on 20 July 2023) | GRID 1 km × 1 km | (1) Used to calculate the ecological benefit enhancement potential $E$: The improvement of multiple ecosystem service capacity of ecological corridor brought by the conversion of agricultural land to ecological land; (2) Used to calculate the economic input cost $R$: to modify the ecological resistance value of agricultural land |

### 2.3.1. Data Sources from Existing Research Results

The data regarding the locations and planning scope of ecological corridors within the study area were obtained from a previously published research article by the authors [34]. In previous research, we employed a developed simulation and evaluation model for determining the appropriate width for constructing urban ecological corridors. This model allowed us to identify the corresponding planning widths for ecological corridors situated in diverse environmental contexts and conditions. The spatial data divided the 66 ecological corridors in the study area into six width ranges, providing clarity on the location, length, as well as the areas of ecological, agricultural, and constructed lands within each corridor. Figure 3 shows the spatial distribution of the 66 ecological corridors and their widths in the Su-Jia-Hu area, and Table 2 demonstrates the six suitable width ranges for ecological corridor construction and detailed corridor information.

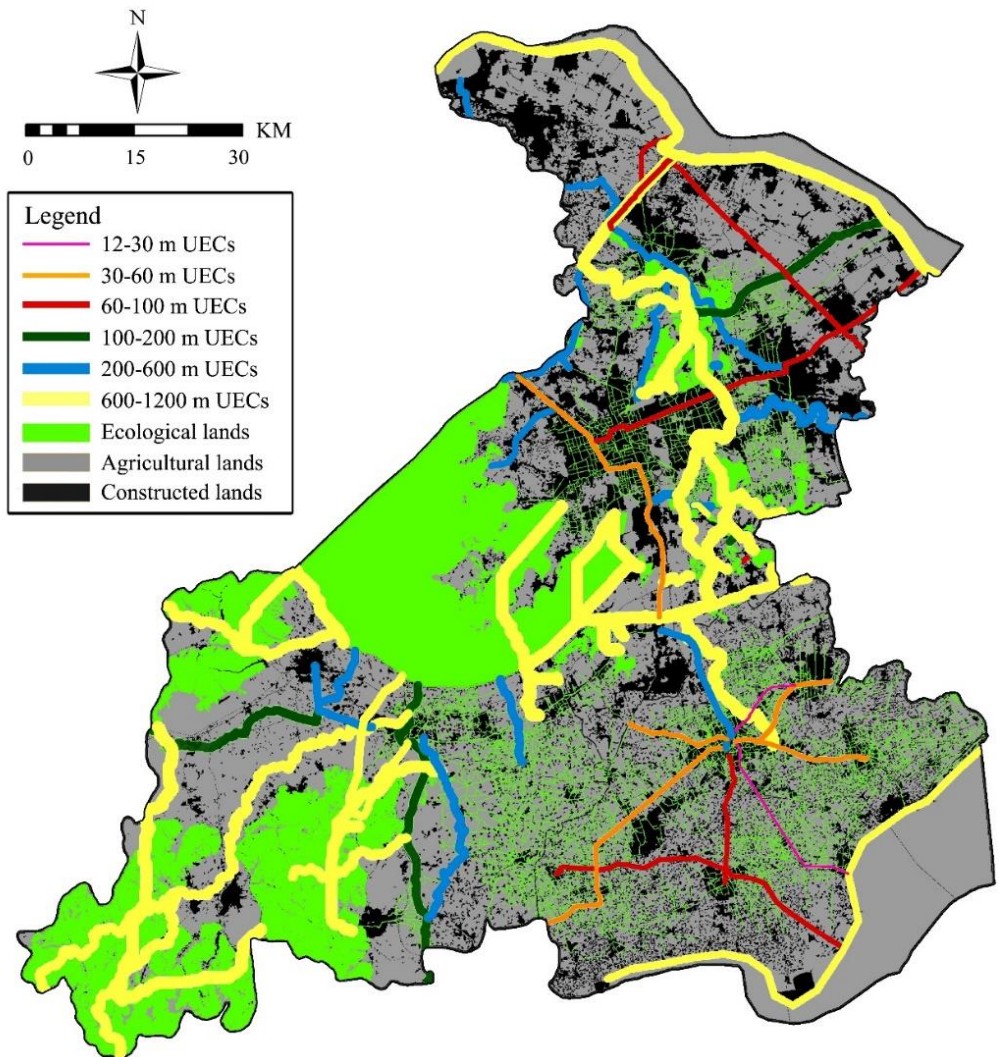

**Figure 3.** Spatial distribution of 66 ecological corridors and their widths in Su-Jia-Hu area (source: [34]).

**Table 2.** The suitable construction width ranges of 66 ecological corridors and detailed corridor information.

| Width Range | Number of UEC | Total Length (km) | Total Area (km²) | Location | Main Types of UECs |
|---|---|---|---|---|---|
| 12–30 m | 2 | 291.3 | 1.5 | Inner city | Existing urban riverside greenways in the planning |
| 30–60 m | 5 | 188.7 | 7.6 | | Existing urban green corridors in the planning |
| 60–100 m | 7 | 254.1 | 17.5 | Intertown | Natural water system corridors, intercity/interval corridors |
| 100–200 m | 4 | 154.9 | 23.4 | Suburban and rural areas | Natural water corridors, suburban and rural road corridors |
| 200–600 m | 13 | 396.8 | 135.0 | | Natural water corridors |
| 600–1200 m | 35 | 1532.5 | 1605.7 | | Potential green corridors, large river corridors |
| Total | 66 | 2818.3 | 1790.6 | — | — |

Source: [34].

This study utilized the data from the author's published research, which included the level assessment of the demand for three ecosystem services in the constructed land of the study area [46]. The three crucial ecosystem services for the study area are flood regulation service, microclimate regulation service, and outdoor recreational service. By applying a natural break method, the assessment results were divided into five levels for each individual ecosystem service demand in the constructed land of the study area. These levels were assigned values ranging from 1 to 5. In this study, we utilized the data to modify the baseline ecological resistance value of the constructed land and calculated the spatial value of the "economic input cost R" using the modified landscape ecological resistance surface. The spatial maps of the three key ecosystem service demands in the constructed lands of the Su-Jia-Hu area are provided in Appendix A.

2.3.2. Other Data Sources

The land use data were generated through supervised classification of Landsat satellite images downloaded from the USGS (http://glovis.usgs.gov/, accessed on 20 July 2023) website. Taking the revised national standard "Current Land Use Classification" (GB/T 21010-2017) [47] organized by the Ministry of Land and Resources as a reference and considering the land use characteristics and discernibility of remote sensing information in the Su-Jia-Hu area, the land use types in the study area were classified into three major categories: agricultural land, constructed land, and ecological land. The classification accuracy for constructed land is above 70%, while the accuracy for other categories is above 80%. The agricultural land in the study area is mainly divided into paddy fields and dry land. The former refers to agricultural land with guaranteed water sources and irrigation facilities and is used to grow aquatic crops such as rice and lotus root. The latter refers to agricultural land that grows crops on natural precipitation and is mainly used to grow vegetables. They are a common type of agricultural land in the plain river network area and also the type with the largest proportion of the total area of the study area (40.1%) of the three land use types. The ecological land in the study area can be subdivided into the following six categories: forest, grassland, river, lake and pond, wetland, and unused land, accounting for 35.5% of the total area of the study area. Constructed land includes urban land, rural residential land, transportation land, industrial land, and other construction land, accounting for 24.4% of the total area of the study area. The constructed land in the study area is distributed in the vast agricultural hinterland in a small, concentrated, and large-scattered pattern.

We utilized the existing spatial distribution dataset of ecosystem service values for terrestrial ecosystems in China (2015) [48] and directly extracted the values of three ecosystem services in the study area's agricultural land, which include hydrological regulation, climate regulation, and landscape aesthetic services. At the level of agricultural land patches, we overlaid and summarized the data of individual ecosystem service values (in yuan per hectare). Based on the characteristics of the total value of the three ecosystem services, we applied the natural break method to divide the data into five categories and assigned values ranging from 1 to 5 accordingly. We transferred the categorical values into ArcGIS and generated the spatial distribution of the three individual ecosystem service values in the agricultural land of the study area (please see Appendix B). We utilized this data to modify the baseline ecological resistance values for agricultural land and conducted spatial calculations of the "economic input cost *R*" using the modified landscape ecological resistance surface.

Furthermore, we assigned the highest level, denoted as 5, to the ecological land's capacity to provide multiple ecosystem services, while the lowest level, denoted as 0, was assigned to the constructed land's capacity to provide multiple ecosystem services. The multiple ecosystem service capacity of agricultural land was represented by the sum of its three individual ecosystem service values. Therefore, this data were also utilized to calculate the "ecological benefit enhancement potential *E*", which referred to the increase in the level

of multiple ecosystem service capacity associated with the conversion of agricultural land into ecological land.

For data sources with inconsistent original resolutions, such as data with a resolution of 1 km × 1 km, we performed a refined interpolation after converting the data from raster to point format in ArcGIS. This conversion resulted in a point grid dataset (.shp) with a resolution of 1 km × 1 km. Subsequently, we utilized the "Trend" within the "Interpolation Analysis" tool in the "Spatial Analyst" extension to interpolate the points into a raster surface with a resolution of 15 m × 15 m.

### 2.4. Correction of Ecological Resistance Value for Non-Ecological Lands

Firstly, we established the baseline ecological resistance values for agricultural land, constructed land, and ecological land based on the existing literature on areas adjacent to the study area [10,49] (Table 3). Following the resistance assignment method utilized by CHEN, et al. [50], we considered ecological patches as the most suitable habitat for species' habitation, reproduction, and migration and, therefore, assigned a constant ecological resistance value of 1 to them. Rivers, on the other hand, act as linear spaces that distinctly differ from the surrounding matrix, impeding species migration and the flow of ecosystem services. Consequently, we assigned an ecological resistance value of 20 to rivers. Then, we utilized the spatial distribution data of the three individual ecosystem service values of agricultural land, as well as the spatial distribution data of the three individual ecosystem service demands for constructed land, to modify the baseline ecological resistance values for agricultural land and constructed land.

**Table 3.** Values of basic landscape ecological resistance of various land uses in Su-Jia-Hu area.

| Land Use Types | Land Use Number | Subclass | Values |
|---|---|---|---|
| Agricultural land | 1 | — | 250 |
| Constructed land | 2 | Rural residential land | 1000 |
| | 3 | Urban land | 2000 |
| Ecological patches | 4 | Forest, grassland, lake and pond, unused land | 1 |
| River corridors | 5 | — | 20 |

From the perspective of ecological suitability, we adjusted the baseline ecological resistance values for agricultural land. We believe that agricultural land with higher ecosystem service values exhibits lower resistance in terms of ecosystem service transmission and delivery compared to other agricultural land. This lower resistance implies lower costs for ecological construction and indicates a smaller investment required for its conversion into ecological land. Therefore, its baseline ecological resistance value should be adjusted to be smaller than that of other agricultural land. Furthermore, we also modified the baseline ecological resistance value for constructed land from the perspective of meeting human objective needs. We believe that constructed land with higher levels of ecological service demand entails a greater necessity to be converted into ecological land to meet and balance local human needs compared with other constructed land. Therefore, their baseline ecological resistance values should be artificially set to be smaller than those of other constructed land, thereby creating or enhancing the potential for ecological system service flow and transfer to these constructed areas.

Following the aforementioned modification principles, the baseline ecological resistance values for "agricultural land", "rural residential land", and "urban land" in Table 3 were modified using the ecological system service value grades of agricultural land and

the ecological system service demand grades of constructed land. The expression for the modified comprehensive ecological resistance value $R'_{ri}$ is as follows:

$$R'_{ri} = \begin{cases} R_{ri} - \sum\limits_{k=1}^{5} (m \times AESV_{irk}), & \text{when } r = 1 \\ R_{ri} - \sum\limits_{k=1}^{5} (n \times CESD_{irk}), & \text{when } r \in [2,3] \end{cases} \quad (1)$$

where $R_{ri}$ represents the baseline ecological resistance value for landscape unit $i$ in the $r$th land use category; $r$ denotes the land use category identification number corresponding to landscape unit $i$. $AESV_{irk}$ represents the grade value of the $k$th ecological system service value for agricultural land unit $i$ when its land use type is agricultural land; $m$ represents the modification coefficient for the ecological resistance value of the agricultural land unit. $CESD_{irk}$ represents the grade value of the $k$th ecological system service demand for constructed land unit $i$ when its land use type is constructed land; $n$ represents the modification coefficient for the ecological resistance value of the constructed land unit. Taking into account that even for constructed land with the highest overall demand level (i.e., when the three individual ecological system service demand grade values $CESD_{irk}$ for the constructed land are all 5), its modified ecological resistance value should still be greater than the baseline ecological resistance value for agricultural land ($R_{ri} = 250$). Through experimentation, we set the modification coefficient $n$ for the ecological resistance value of constructed land to 30. Moreover, even for agricultural land with the highest ecological system service value (i.e., when the three individual ecological system service value grade values $AESV_{irk}$ for the agricultural land are all 5), its modified ecological resistance value should still be greater than the baseline ecological resistance value for rivers ($R_{ri} = 20$). Based on experimentation, the modification coefficient m for the ecological system service value of agricultural land was set to 4. Using Equation (1), we applied the modification to the baseline ecological resistance values for individual agricultural land patches and constructed land patches within the study area, resulting in the modified ecological resistance surface spatial data used for calculating the "economic input cost R".

## 2.5. Comprehensive Benefit Index of Ecological Corridors

To comprehensively assess the benefits of ecological corridors in ecological, social, and economic aspects within the metropolitan area, this study designed and constructed the Comprehensive Ecological Corridor Benefit Index *I*. It was quantified as a function of "ecological benefit enhancement potential *E*, "economic input cost *R*", and "social coordination cost *S*", represented as *I*= f($E, R, S$). The specific form of the function is as follows:

$$I = K - \frac{k_1}{k_3 + E^2} - k_2 R * S \quad (2)$$

where *E* represents the potential for ecological benefits enhancement, indicating the increase in the capacity of the ecological corridor to provide multiple ecosystem services when non-ecological lands are converted into ecological lands and are positively correlated with the *I*. *R* represents the economic input cost, which was determined by the overall ecological resistance value for non-ecological lands that need to be converted into ecological lands; r is negatively associated with *I*. *S* denotes the social coordination cost, with the total population residing in the non-ecological lands that need to be converted into ecological lands within the ecological corridor serving as a proxy; *S* is negatively correlated with *I*. $k_1$, $k_2$ and $k_3$ are constants, and they are set so that the impact of *E*, *R*, and *S* on *I* is of the same magnitude because here we assumed that the impact of ecological, social, and economic aspects on the comprehensive benefits of ecological corridors is equivalent. Considering

the value ranges of *E*, *R*, and *S*, we set $k_1 = 10{,}000{,}000$, $k_2 = 0.0000000001$, and $k_3 = 0.1$. The specific expression of *K* is as follows:

$$K = \gamma * \max\left(\frac{k_1}{k_3 + E^2} + k_2 R * S\right) \tag{3}$$

where $\gamma$ is a scaling factor, with $\gamma > 1$. The evaluation model for the Comprehensive Ecological Corridor Benefit Index is based on the following assumptions: a significant increase in the capacity of multiple ecosystem services resulting from the conversion of non-ecological to ecological lands within the ecological corridor, a lower overall ecological resistance to overcome during the conversion and construction process, and a minimal need for coordination or influence on the social population originally involved in the area. Under these conditions, the land conversion within the ecological corridor yields the maximum comprehensive benefits, providing a feasible solution that balances ecological, social, and economic dimensions.

To further clarify the specific land conversion within the ecological corridor that maximizes its comprehensive benefits, we established a functional relationship between the converted agricultural land area *A*, constructed land area *C*, and the Comprehensive Ecological Corridor Benefit Index $I = h(A, C)$. We calculated *E* by multiplying the average level of multiple ecosystem service improvement per unit area of converted agricultural land or constructed land within the ecological corridor by the ratio *A/C*. *R* was calculated by multiplying the modified ecological resistance value per unit area of converted agricultural land or constructed land within the ecological corridor by the ratio *A/C*. *S* was calculated by multiplying the population density of the study area by the ratio *A/C*. By substituting the above indicators into Equation (2) and performing the necessary expansion, we obtained the specific functional expression of *I* in terms of *A* and *C*.

$$I_i = K - \frac{1}{aA_i{}^2 + bA_iC_i + dC_i{}^2} - eA_i{}^2 - mA_iC_i - nC_i{}^2 \tag{4}$$

where *i* represents the identification number of the ecological corridor within the study area. $A_i$ and $C_i$ represent the area of converted agricultural land and constructed land, respectively, within ecological corridor *i*, and $A_i \in (0, x_i)$, $C_i \in (0, y_i)$, where *x* represents the total area of agricultural land within corridor *i*, while *y* represents the total area of constructed land within corridor *i*. The parameters *a*, *b*, *d*, *e*, *m*, and *n* were derived from the relevant calculation data and obtained by substituting them into the equation and performing the necessary expansion (please see Appendix C for details). Using Matlab 6.5 software, we determined the values of *A* and *C* within their respective ranges that maximize the value of *I* in Equation (4). The ratios *A/x* (%) and *C/y* (%) were defined as the optimal conversion ratios of agricultural land and constructed land, respectively, that maximize the comprehensive benefits of the ecological corridor. This provides quantitative reference recommendations for optimizing land use structure within the ecological corridor.

## 3. Results

### 3.1. Optimal Conversion Ratio of Agricultural Land in Ecological Corridors and Analysis

Using the evaluation model for the Comprehensive Ecological Corridor Benefit Index, we calculated the optimal conversion ratios of agricultural land within the planning scope of 66 ecological corridors identified in the study area. Detailed results of agricultural land area, optimal conversion area, and ratio for each corridor are provided in Appendix D.

We conducted a Pearson's correlation analysis using the correlation analysis tool in PASW Statistics 18 software to examine the statistical relationship between ecological corridor area, agricultural land area within the corridor, optimal conversion area for agricultural land, and conversion ratio. The results indicate that as the planned area of the ecological corridor increases, the corresponding agricultural land area within the corridor also increases. However, the optimal conversion ratio of agricultural land to ecological

space decreases accordingly. Conversely, when the planned area is smaller, the optimal conversion ratio becomes larger. In other words, a significant negative correlation exists between the optimal conversion ratio of agricultural land in the ecological corridor and the corridor area, total agricultural land area, and optimal conversion area of agricultural land (please see Appendix E).

Based on the above results, we analyzed and proposed the overall optimization principle for the conversion of agricultural land within the ecological corridor. When the agricultural land within the corridor is of substantial scale, the ratio of converting these lands into ecological lands should be relatively small. This approach aims to maintain the original scale and spatial pattern of agricultural land within the corridor, which is beneficial for preserving the consistency between habitat ecological structure and species within the habitat [28] and enhances the functional continuity of land use within the ecological corridor. When the scale of agricultural land within the ecological corridor is small, it indicates that the dominant land use types in the corridor are ecological land or constructed land. In the former scenario, the spatial pattern of agricultural land often tends to be scattered and fragmented. Therefore, it is advisable to convert it into corresponding dominant ecological land, thereby ensuring functional and structural consistency within the habitats of the ecological corridor. In the latter scenario, increasing the ratio of agricultural land converted into ecological land at suitable locations helps to compensate for the lower ecological quality resulting from excessive development within the ecological corridor. However, it is important to ensure that all conversions of agricultural land are carried out while protecting the fundamental agricultural land area.

To further investigate the optimal conversion of agricultural land within the ecological corridor, we utilized the spatial clustering analysis tool in ArcGIS. The ecological corridors, characterized by different optimal conversion ratios for agricultural land, were classified into three categories: (1) low conversion level as agricultural land with an optimal conversion ratio ranging from 0% to 35%; (2) medium conversion level as agricultural land with an optimal conversion ratio ranging from 35% to 77%; and (3) high conversion level as agricultural land with an optimal conversion ratio ranging from 77% to 100%. Based on the analysis of the number of ecological corridors, optimal conversion ratios of agricultural land, and their respective areas within each conversion level, we summarized the corresponding optimization recommendations (Table 4). Figure 4 presents the spatial distribution of ecological corridors with different optimal conversion levels for agricultural land in the study area.

**Table 4.** Statistics of the optimal conversion ratio of different degrees of agricultural lands in the ecological corridors and corresponding optimization suggestions.

| Conversion Degree | Conversion Ratio Range | Optimal Conversion Area of Agricultural Land (km$^2$) | Number of Corridors Involved | Conversion and Optimization Recommendations |
|---|---|---|---|---|
| Low | 0%~35% | 166.95 | 23 | Layout: select agricultural land in local key locations for conversion, such as agricultural land in corridors that interrupt the ecological spatial continuity or scattered and independent agricultural land for priority conversion |
| | | | | Type: to be converted to the type of ecological land that predominates within the corridor or the type of ecological land that is in close proximity to the converted agricultural land |

**Table 4.** *Cont.*

| Conversion Degree | Conversion Ratio Range | Optimal Conversion Area of Agricultural Land (km²) | Number of Corridors Involved | Conversion and Optimization Recommendations |
|---|---|---|---|---|
| Moderate | 35%~77% | 127.16 | 17 | Layout: select agricultural land at the edge of the corridor, patches of agricultural land scattered inside the corridor, or agricultural land that is located in an ecological land that presents a stepping-stone layout for priority conversion |
| | | | | Type: to be converted to the type of ecological land that aligns with the dominant ecological land or ecological stepping stones |
| High | 77%~100% | 151.38 | 26 | Layout: select agricultural land at the edge of the corridor or agricultural land that is within the narrower corridor and lacks an ecological base for priority conversion |
| | | | | Type: to be converted to the type of ecological land that predominates within a corridor or is in close proximity to the converted agricultural land or to the ecological land type with a high capacity to provide multiple ecosystem services (e.g., forests) |
| Total | | 445.49 | 66 | — |

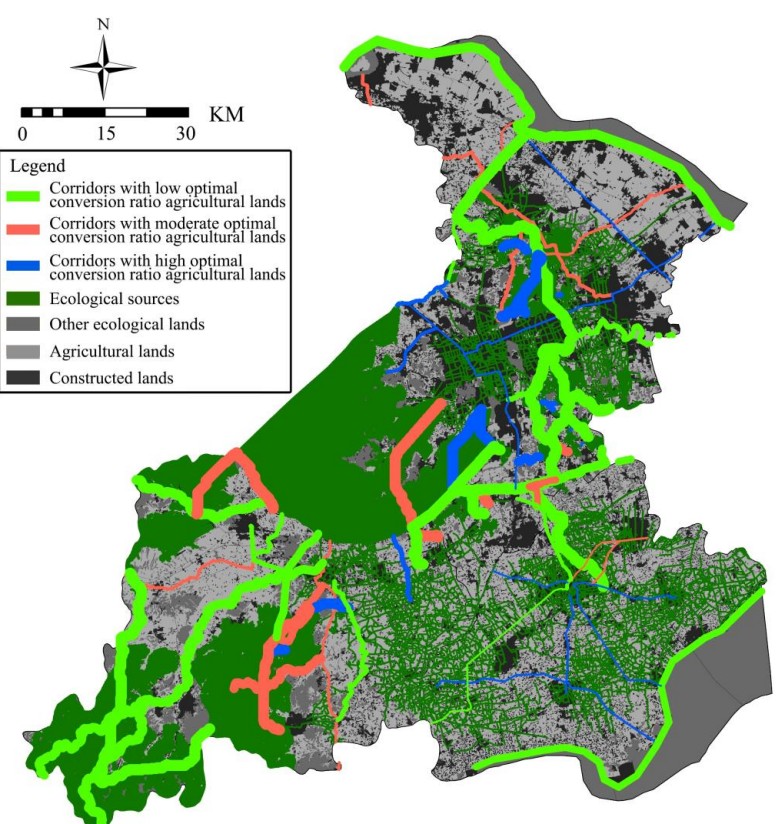

**Figure 4.** Spatial distributions of ecological corridors, including different optimal conversion ratios of agricultural lands in the study area (source: drawn by the authors).

We also depicted the distribution of agricultural land within ecological corridors with different optimal conversion levels (Figure 5). Figure 5a showcases that ecological corridors with a low level of optimal conversion ratio for agricultural land (0%~35%) are characterized by longer lengths, larger planning widths, and broader coverage. Additionally, they are associated with a greater number of high-quality ecological source areas. Within these ecological corridors, most agricultural land areas are sizable, spatially contiguous, and exhibit good spatial connectivity. Agricultural land itself also has the ability to provide multiple ecosystem services, and maintaining consistency between habitat types, ecological structure, and species within ecological corridors facilitates the flow of biodiversity among habitats. Therefore, it is recommended to convert agricultural land into ecological land in these corridors at a relatively low ratio. This implies that only small-scale conversions should be conducted on agricultural land located in critical local positions, converting them into dominant ecological land types within the ecological corridor or adjacent to the converted agricultural land. For example, agricultural land that interrupts the continuity of ecological lands within the corridor or scattered and isolated agricultural land patches (as illustrated in Figure 6).

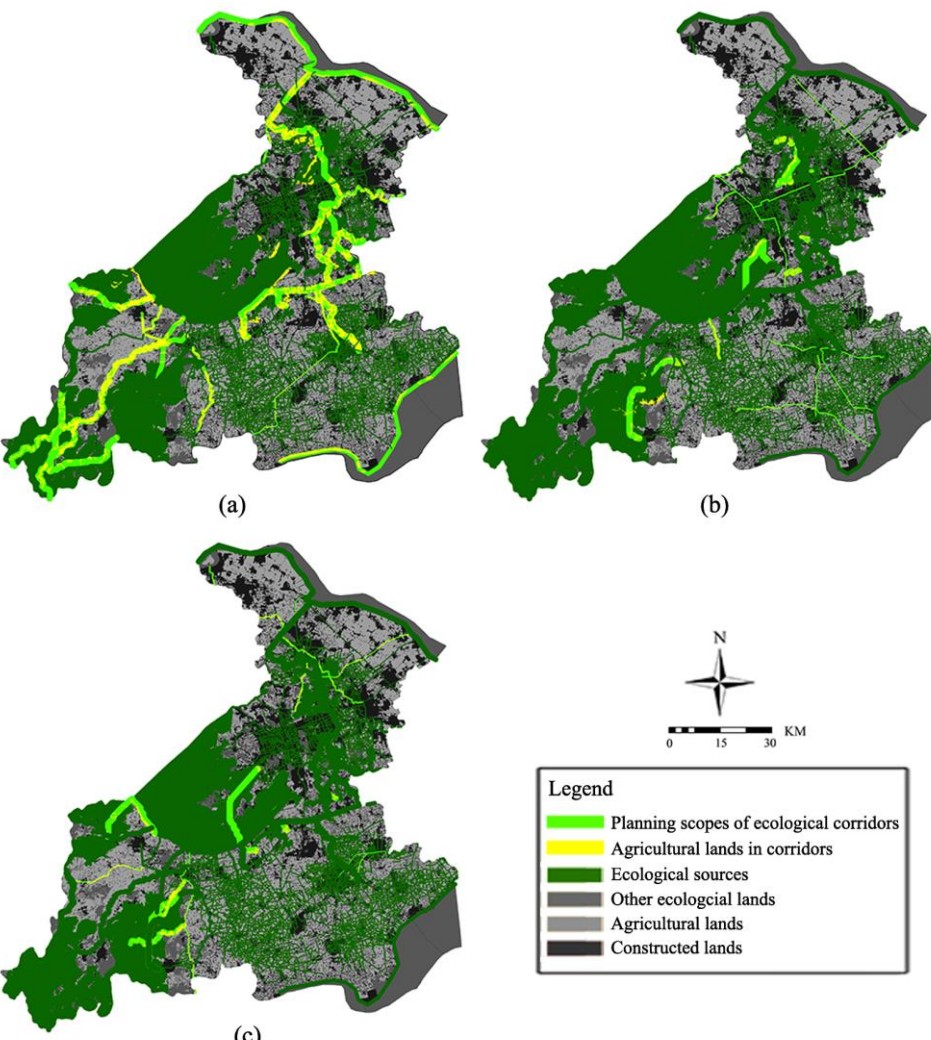

**Figure 5.** Spatial distributions of agricultural lands in the ecological corridors with different optimal conversion ratios. Spatial distributions of agricultural lands with: (**a**) low optimal conversion ratio, (**b**) moderate optimal conversion ratio, and (**c**) high optimal conversion ratio in the ecological corridors. (source: drawn by the authors).

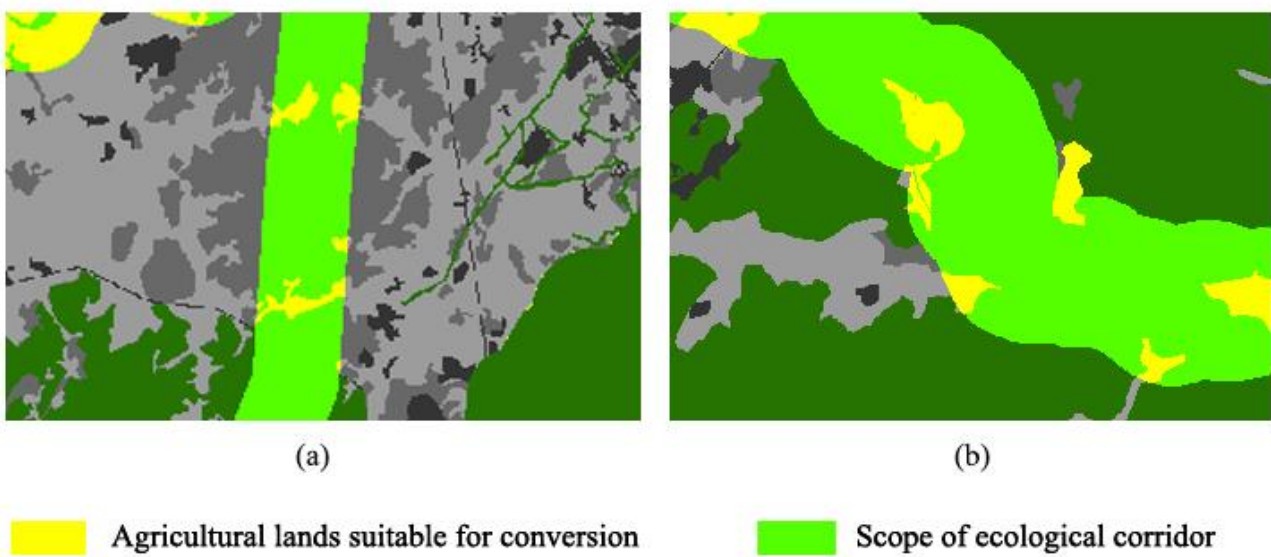

**Figure 6.** Schematic diagram of optimization proposals for agricultural lands with low optimal conversion ratio: (**a**) Agricultural land that interrupts the continuity of ecological space within the ecological corridor; (**b**) Scattered small-scale agricultural lands within the ecological corridor (source: drawn by the authors using ArcGIS 10.6 software).

Ecological corridors with a high level of optimal conversion ratio for agricultural land (77%~100%) can be divided into two scenarios (as shown in Figure 5b): One scenario consists of shorter corridors with wider planning width, which connect multiple ecological source areas. Examples of such corridors can be found in several locations, including the central part of Suzhou, the eastern side of Taihu Lake, and the central part of Huzhou. In these aforementioned ecological corridors, ecological land is the dominant spatial type, while agricultural land entails a small area, scattered distribution, and poor connectivity. This increases the heterogeneity in spatial types and structures within the corridors. For these relatively small agricultural lands, it is recommended to convert them into ecological land at a larger ratio, particularly those located at the edges of the planned scope of the corridor (as shown in Figure 7a). The rationale is that their presence reduces the width of the ecological corridor and increases the edge effect, which affects the ecological functionality of the corridor [51]. The other case involves longer but narrower ecological corridors that pass-through regions with limited ecological land bases or densely populated constructed areas. Examples include the ecological corridors in the northern part of Suzhou and the southern part of Jiaxing, as well as several corridors that traverse through the urban areas of Suzhou and Jiaxing. In these ecological corridors, the spatial continuity of agricultural land is compromised by the presence of constructed areas. Additionally, these corridors have limited existing ecological land and lack a robust ecological base. Therefore, it is highly recommended to convert a significant ratio of agricultural land into ecological lands that provide a high level of multiple ecosystem services, such as shelter forests or riparian forest belts with a woody-herbaceous structure. This conversion would increase the vegetation complexity within the corridor and enhance the quality of the ecological corridor [52] (please see Figure 7b).

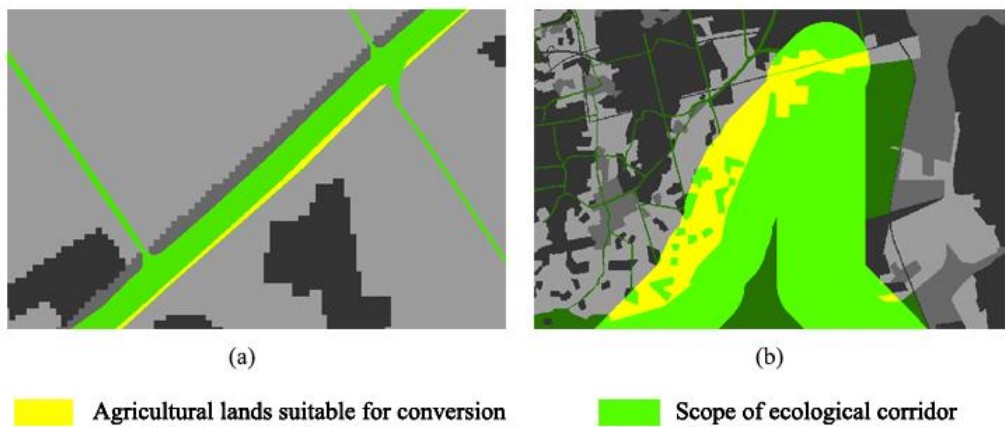

**Figure 7.** Schematic diagram of optimization proposals for agricultural land with a high optimal conversion ratio: (**a**) Agricultural land with small area within the planning scope of narrower corridors; (**b**) Agricultural land located at the edge of the ecological corridor enhances the edge effect (source: drawn by the authors using ArcGIS 10.6 software).

Figure 5c presents the ecological corridors with a medium level of optimal conversion ratio for agricultural land (35%~77%) and the distribution of agricultural land within these corridors. Within these ecological corridors, there are both agriculturally continuous areas and fragmented patches of agricultural land. For these agricultural lands, it is recommended to prioritize the conversion of the following types of agricultural lands based on the optimal conversion ratio: agricultural lands located at the edges of the corridors, scattered patches of agricultural land within the corridors, and agricultural lands located between ecologically significant spaces that form a stepping-stone network. This approach aims to establish spatial connectivity by connecting existing ecological lands, as illustrated in Figure 8.

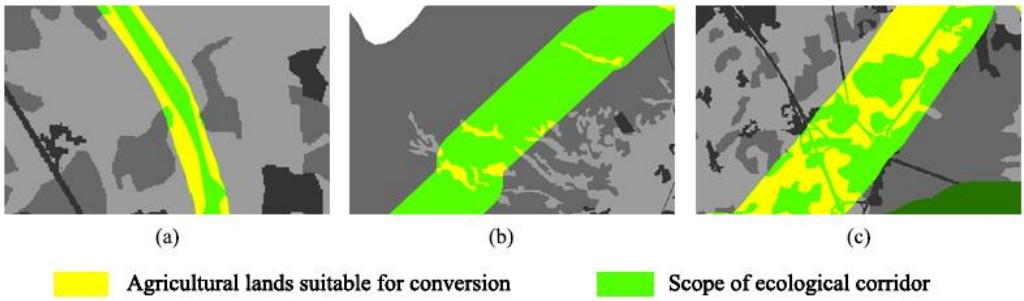

**Figure 8.** Schematic diagram of optimization proposals for agricultural land with moderate optimal conversion ratio: (**a**) Agricultural land located at the edge of the ecological corridor enhances the edge effect; (**b**) Scattered small-scale agricultural lands within the ecological corridor; (**c**) Agricultural land between ecological stepping stones (source: drawn by the authors using ArcGIS 10.6 software).

### 3.2. Optimal Conversion Ratio of Constructed Land in Ecological Corridors and Analysis

Using the evaluation model for the Comprehensive Ecological Corridor Benefit Index, we calculated the optimal conversion ratios for constructed land within the planning scope of 66 ecological corridors in the study area. Detailed results and statistics of the constructed land area, optimal conversion area, and their ratios for each corridor can be found in Appendix F.

We conducted a Pearson's correlation analysis using the correlation analysis tool in SPSS 29 software to examine the statistical and calculated results of ecological corridor area, constructed land area within the corridor, optimal conversion area, and ratio for constructed land. The results indicate that as the ecological corridor area increases, the area of constructed land within the corridor also increases, while the optimal conversion

ratio of constructed land to ecological land decreases. Conversely, when the corridor area decreases, the optimal conversion ratio increases. In other words, a significant negative correlation exists between the optimal conversion ratio of constructed land within the ecological corridor and both the corridor area and the area of constructed land within the corridor (please see Appendix G). This pattern is consistent with the observed trend in the optimal conversion ratio of agricultural land. Additionally, we also observed a significant positive correlation between the optimal conversion ratio for constructed land and that for agricultural land (please see Table 5).

**Table 5.** Pearson's correlation analysis results in optimal conversion ratio of agricultural land and constructed land.

| | | Optimal Conversion Ratio of Agricultural Land | Optimal Conversion Ratio of Constructed Land |
|---|---|---|---|
| Optimal conversion ratio of agricultural land | Pearson's correlation | 1 | 0.587 ** |
| | Significance (two sides) | | 0.000 |
| | N | 66 | 66 |
| Optimal conversion ratio of constructed land | Pearson's correlation | 0.587 ** | 1 |
| | Significance (two sides) | 0.000 | |
| | N | 66 | 66 |

** Significantly correlated at the level of 0.01 (two-sided test).

To further investigate the optimal conversion of constructed land within the ecological corridors, we employed the spatial clustering analysis tool in ArcGIS to classify the corridors into three categories based on their optimal conversion ratios of constructed land: (1) low conversion level as optimal conversion ratio of constructed land at 0%~37%; (2) medium conversion level as optimal conversion ratio of constructed land at 37%~87%; and (3) high conversion level as optimal conversion ratio of constructed land at 87%~100%. We conducted a statistical analysis on the number of ecological corridors and the optimal conversion ratios and areas of constructed land within each conversion level. Based on the results, we have summarized the corresponding optimization recommendations in Table 6. Figure 9 illustrates the spatial distribution of ecological corridors with different levels of optimal conversion of constructed land in the study area.

We have also mapped the distribution of constructed land within ecological corridors with different levels of optimal conversion of constructed land (Figure 10). As shown in Figure 10a, ecological corridors with a low level of optimal conversion ratio for constructed land (0%~37%) are predominantly characterized by longer lengths, wider planning widths, and larger spatial scopes. These corridors also encompass a substantial overall scale of constructed lands and are distributed in contiguous patches. In this scenario, a relatively small ratio of constructed land should be converted into ecological land. This is because large and contiguous human settlements have their own integrity as cultural–ecological systems. This implies that, on the one hand, forcibly converting urban constructed land into ecological land incurs high economic costs and entails social infeasibility due to the involvement of numerous stakeholders. On the other hand, forcefully converting constructed land located within a larger extent of surrounding constructed land into ecological land would significantly compromise its ecological quality due to excessive human disturbances. Therefore, for ecological corridors that pass through urban areas with a substantial amount of constructed land, it is more appropriate to focus on enhancing the ecological quality and spatial continuity within the existing constructed land by incorporating green infrastructure. This can be achieved through implementing measures such as neighborhood parks, road greening, and rain gardens, as well as vertical greening such as rooftop gardens and green walls (Figure 11a). On the other hand, for ecological corridors that traverse rural settlements and encompass a significant amount of constructed land, opportunities can

be found in the context of the "Beautiful Countryside" and "New Socialist Countryside" initiatives, where village greening projects with a focus on improving the surroundings of homes and villages can be extensively promoted [53] (Figure 11b). Furthermore, there are also cases where constructed land within the ecological corridor forms a stepping-stone pattern. For ecological corridors that include such constructed land, it is advisable to prioritize the conversion of those constructed areas that are most likely to disrupt the flow of biodiversity or ecological services within the corridor. This selection should be based on the recommended optimal conversion ratio for constructed land (Figure 10c).

**Table 6.** Statistics of the optimal conversion ratio of different degrees of constructed lands in the ecological corridors and corresponding optimization suggestions.

| Conversion Degree | Conversion Ratio Range | Optimal Conversion Area of Constructed Land (km²) | Number of Corridors Involved | Conversion and Optimization Recommendations |
|---|---|---|---|---|
| Low | 0%~37% | 37.58 | 21 | Layout: Add green infrastructure on the basis of the urban constructed land located in the ecological corridor, and carry out the greening construction of the village bay in combination with the residence in the constructed land of rural settlements. Select the constructed lands that are most likely to block the flow of organisms or ecosystem services in the ecological corridor for priority conversion. |
| | | | | Type: Construct green infrastructures such as the block park, road greening, and rain garden, as well as vertical three-dimensional greening such as roof greening, wall demolition, and greening. To be converted to the type of ecological land that dominates within the ecological corridor or the type of ecological land that is in close proximity to the converted constructed land. |
| Moderate | 37%~87% | 36.13 | 10 | Layout: Select small-scale constructed lands located within large-scale ecological lands, constructed lands most likely to hinder the flow of biological and ecosystem services, and constructed lands located in buffer zones on both sides of narrow river corridors that belong to small human settlements for priority conversion. Select constructed lands that are clustered or connected in ecological corridors and belong to medium or large urban areas for green infrastructure construction. |
| | | | | Type: To be converted to the type of ecological land that dominates in the ecological corridor or is in close proximity to the converted constructed lands. Construct small-scale urban green infrastructures such as community green spaces, pocket parks, and rain gardens, as well as green planting, ecological slope protection, or green belts along rivers. |
| High | 87%~100% | 79.37 | 35 | Layout: Select scattered and small-scale constructed land located inside the large-scale ecological land in the ecological corridor for priority conversion, constructed lands in river corridors that belong to small-scale settlements, constructed lands located in an ecological corridor that belongs to a medium-sized urban area, and constructed lands located within a narrow river corridor that belongs to a large-scale urban area for priority conversion |
| | | | | Type: To be converted to the type of ecological land that dominates in the ecological corridor or is in close proximity to the converted constructed land. Construct small-scale urban green infrastructures such as community green spaces, pocket parks, and rain gardens, increase planting along rivers, and build ecological slope protection or green belts along rivers. |
| Total | | 153.08 | 66 | — |

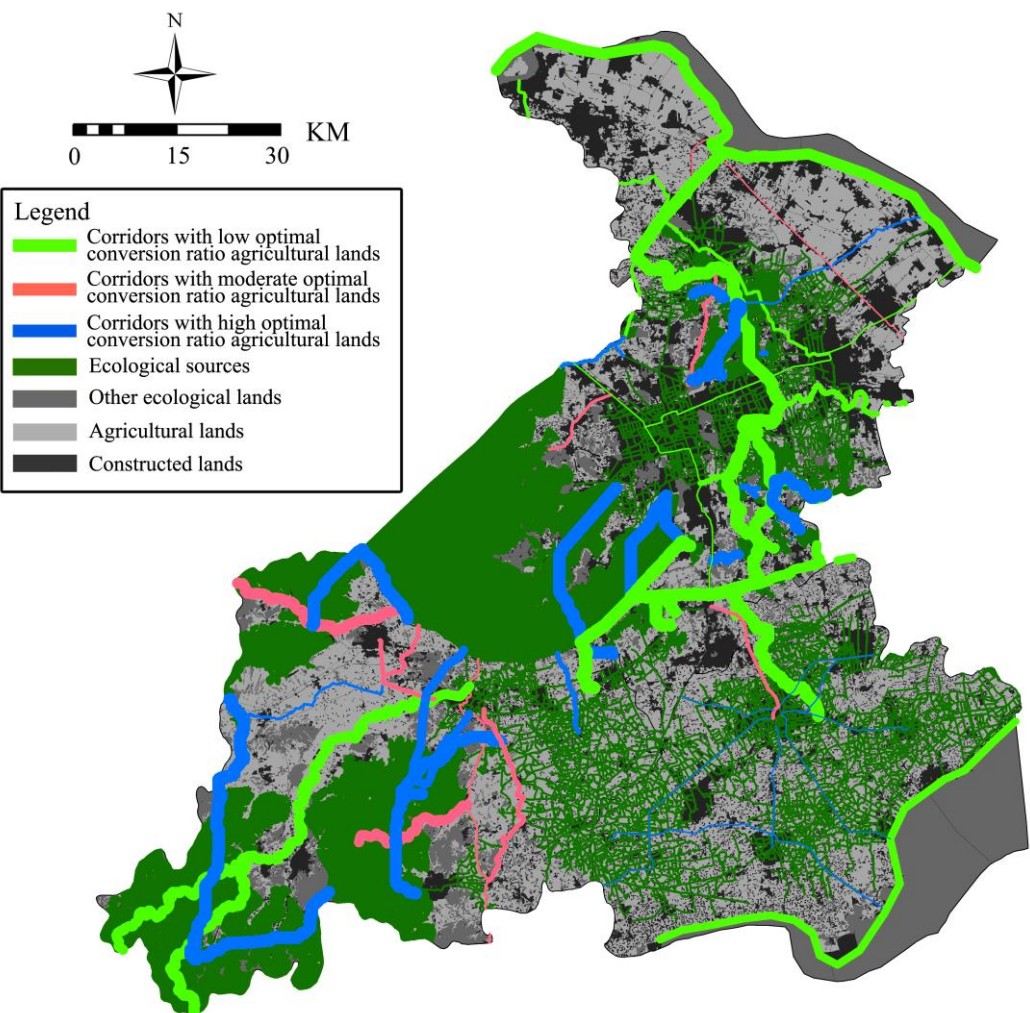

**Figure 9.** Spatial distributions of ecological corridors, including different optimal conversion ratios of constructed lands in the study area (source: drawn by the authors).

Ecological corridors with a high level of optimal conversion ratio for constructed land (87%~100%) can be classified into two scenarios (Figure 10b): Firstly, there are long corridors with wide planning widths that connect multiple large-scale ecological source areas. Examples include several ecological corridors on the eastern side of Lake Taihu and the northern, central, and southern parts of Huzhou City. Within the aforementioned ecological corridors, the constructed lands exhibit a small area, limited scale, and scattered distribution. For such constructed lands, one approach to consider is relocating the existing residents outside the planning scope of the ecological corridor. This would allow for the restoration of the constructed land to predominantly or entirely ecological spaces such as forests, grasslands, and rivers, thereby reducing ecological barriers within the corridor (Figure 12a). Another type of ecological corridor is characterized by a long length but narrow planning width, passing through urban areas or areas with limited ecological land base. Examples include the northern part of Suzhou, the southern part of Jiaxing, and several ecological corridors radiating from the main urban area of Jiaxing. In these ecological corridors, while the overall area of constructed land is relatively small, it is concentrated and continuous, forming an integral part of the external human settlement environment. Therefore, different planning approaches can be adopted based on specific circumstances. For constructed lands located within buffer zones along river corridors that pass through small-scale human settlements, a localized relocation strategy can be implemented to move them outside the scope of the ecological corridor, thereby ensuring the integrity of the river

corridor (as shown in Figure 12b). For constructed lands within the interior of ecological corridors passing through medium-sized urban areas, the strategy involves incorporating small-scale urban green infrastructure such as community green spaces, pocket parks, and rain gardens along the path of the ecological corridor. This approach creates an urban green corridor formed by urban green spaces, establishing an ecological development axis (as depicted in Figure 12c). For constructed lands within narrow buffer zones along river corridors that pass through large-scale urban areas, additional measures can be implemented on the existing spaces, such as planting vegetation along the riverside, constructing ecological slopes, or establishing riverside green belts (as illustrated in Figure 12d).

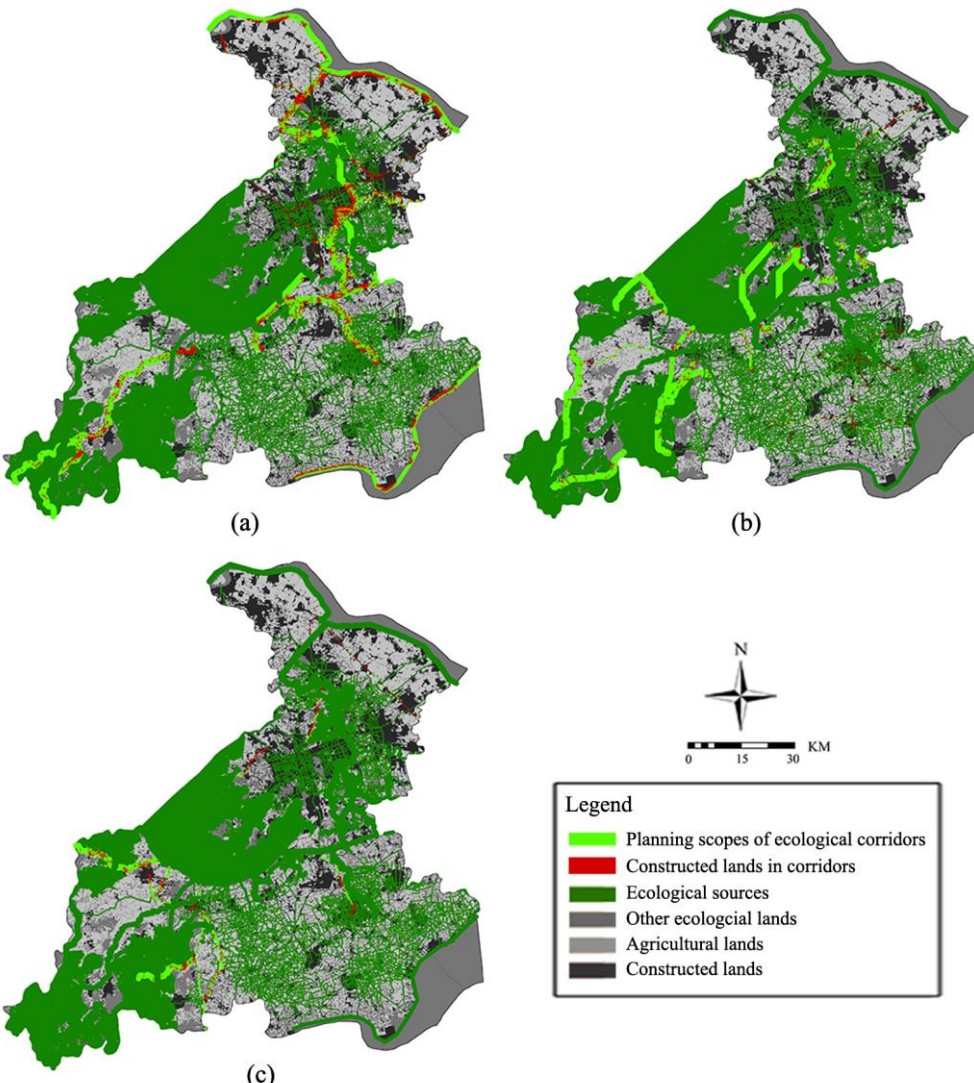

**Figure 10.** Spatial distributions of constructed lands in the ecological corridors with different optimal conversion ratios Spatial distributions of constructed lands with: (**a**) low optimal conversion ratio, (**b**) moderate optimal conversion ratio, and (**c**) high optimal conversion ratio in the ecological corridors. (source: drawn by the authors).

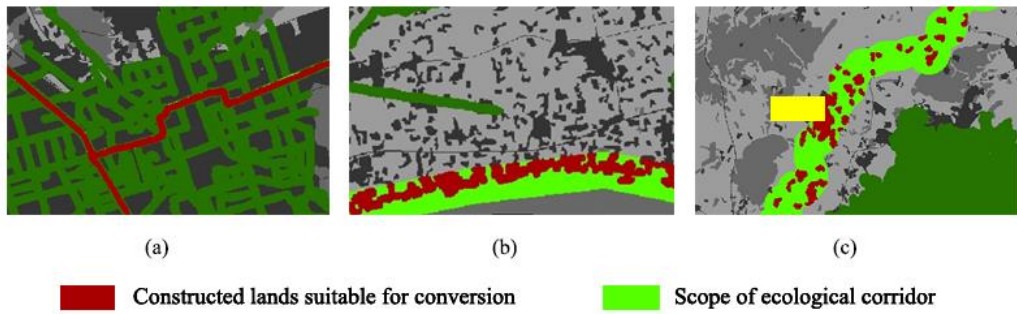

**Figure 11.** Schematic diagram of optimization proposals for constructed lands with low optimal conversion ratios: (**a**) Constructed land located in the ecological corridor passing through the large-scale urban space; (**b**) Constructed land located in the ecological corridor passing through the contiguous rural settlements; (**c**) Constructed land dispersed in ecological corridors but most likely to block biological flow and landscape flow (source: drawn by the authors using ArcGIS 10.6 software).

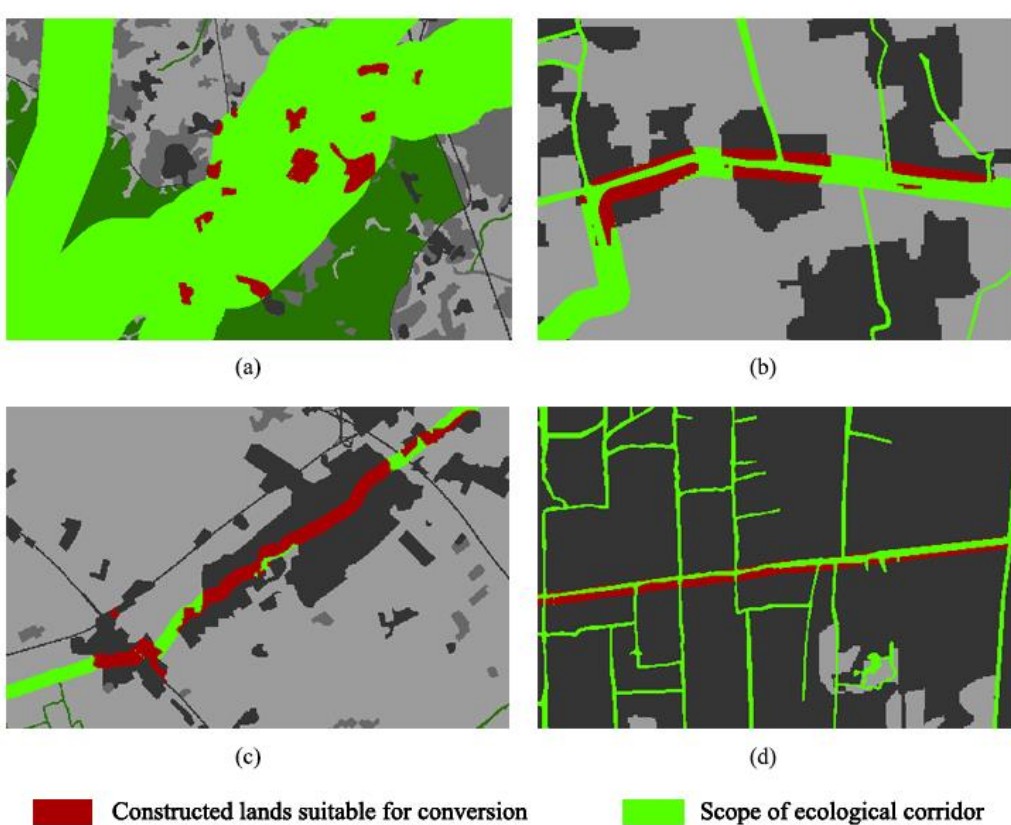

**Figure 12.** Schematic diagram of optimization proposals for constructed lands with high optimal conversion ratio: (**a**) Scattered and small-scale constructed land within large-scale ecological spaces in the ecological corridor; (**b**) Constructed land within buffer zones along the banks of river corridors that pass through small human settlements; (**c**) Constructed land located inside the ecological corridor through the medium volume town spaces; (**d**) Constructed land located within the buffer zone of the river corridor through the large-scale urban spaces (source: drawn by the authors using ArcGIS 10.6 software).

Figure 10c presents ecological corridors with a medium level of optimal conversion ratio for constructed land (37%~87%) and the distribution of constructed land within the corridor. These ecological corridors exhibit both contiguous patches of constructed land and scattered, isolated pockets of constructed land. For these constructed lands, it is advisable to prioritize, taking into account the optimal conversion ratios, the conversion of small-scale constructed lands located within large-scale ecological lands (as shown in

Figure 12a), constructed lands situated in locations that are most likely to impede the flow of biodiversity, landscape, and ecosystem services (as depicted in Figure 11c), and constructed lands within narrow buffer zones along river corridors that are associated with small human settlements (as illustrated in Figure 12b), and convert them into ecological lands. For constructed lands that are contiguous or clustered within ecological corridors and are associated with medium or large urban areas (as depicted in Figure 11a,b and Figure 12c,d), it is recommended to consider the optimal conversion ratios and develop urban green infrastructure of appropriate sizes within the existing constructed lands. This approach will facilitate the flow of ecosystem services within urban regions. Overall, in the process of considering converting constructed lands to ecological lands or adding new green infrastructure within constructed lands, it is necessary to further expand the analysis based on the potential analysis of ecological lands within the urban area [53] or the suitability analysis of urban green space construction [54,55].

## 4. Discussion

### 4.1. Significance of Optimizing Hybrid Landscape Land Structure in Ecological Corridors from the Perspective of Comprehensive Benefits

In rapidly urbanizing areas, the planning and implementation of landscape ecological networks are constrained by direct costs such as limited funding and land availability [56,57]. Therefore, converting non-ecological into ecological lands into built landscape ecological networks entails opportunity costs [58,59], particularly the potential loss of economic development opportunities. It can be said that in metropolitan areas, there is fierce competition for resources, particularly funding and land, in the construction of landscape ecological networks [60]. From an ecological perspective, it is advantageous to have a higher ratio of ecological land within the planned width of an ecological corridor. This enhances the corridor's capacity to provide multiple ecosystem services [34] and improves the internal connectivity within the corridor, thereby increasing the efficiency of spatial flow and transfer of ecosystem services within it. However, purely pursuing maximum ecological benefits while disregarding social needs and economic development stages would significantly increase the economic costs associated with planning and constructing ecological corridors and the overall landscape ecological network [28]. It will also face challenges in garnering social support and political decision-making protection due to competing for resources with other projects [61–63].

A regional landscape ecological network should be seen as an integrative tool for sustainable development rather than just a specific and rigid spatial form objective [64]. Its value lies in connecting and integrating knowledge and approaches from various dimensions, including science, society, economics, politics, and more, to address ecological issues [65]. This allows for the long-term integration of ecological conservation into the development processes of other disciplines [66]. Therefore, the genuinely scientific process of constructing a landscape ecological network is not about identifying an exact spatial network based on conventional models of landscape ecological networks and strictly planning according to that spatial framework. Instead, it involves landscape planners coordinating the relationships between stakeholders, land, and the landscape ecological network based on scientific models within that framework [65]. It can be stated that the responsibility of landscape planners extends beyond identifying the elements, layout, scale of landscape ecological networks, and corresponding protective measures. Their role also involves facilitating dialogue and interplay among complex interest groups within society [65], thus integrating ecological, social, and economic functions throughout the landscape ecological planning process. This integration serves to maximize comprehensive benefits across multiple dimensions, such as "ecological conservation", "social harmony", and "economic development" [67]. This also implies that a landscape ecological network that balances ecological, social, and economic objectives is a "socio-economic feasible solution" achieved through coordination and compromises, instead of the "ecological technically optimal solution". Although this landscape ecological network may not achieve

the highest level of individual attribute optimization, it represents an optimized solution that maximizes the comprehensive benefits in terms of "ecology–society–economy" [68]. This is also why this study adopts a comprehensive benefits perspective to adjust and optimize the hybrid landscape and land–use structure within ecological corridors in rapidly urbanizing areas. This method and model provide new insights for the structural design of ecological corridors in metropolitan areas.

Furthermore, for the construction of landscape ecological networks in metropolitan areas, we believe that moderate economic development does not necessarily lead to environmental degradation. Instead, it enhances the economic value added to landscape ecological networks. For example, stimulating economic development in surrounding areas, increasing real estate value, creating tourism opportunities, and generating employment opportunities [69–71]. Meanwhile, human well-being also relies on the enhancement of the social value of landscape ecological networks. For example, providing recreational and socializing spaces, promoting the physical and mental health of residents, enhancing street vitality and walkability, and fostering a sense of local identity [72–74]. Therefore, while maintaining the ecological sustainability baseline, it is necessary to allocate appropriate space for economic development in the process of ecological construction [75]. Furthermore, the selection of landscape ecological network planning and optimization strategies with the highest social acceptance should be prioritized [76] to achieve maximum feasibility in both social and economic dimensions, thereby allowing for the construction of a landscape ecological network that maximizes comprehensive benefits.

*4.2. Limitations*

The methods and new indicators provided in this study contribute to determining the optimal conversion ratio of agricultural land and constructed land to ecological land within the ecological corridor. However, this analysis only focused on the quantitative aspect and lacked specific spatial guidance, particularly in identifying the priority areas for the conversion of agricultural land or constructed land. While we provided land conversion recommendations for the study area as a practical case in our analysis, it is important to note that these recommendations are not explicitly included in the research methodology and, therefore, may not be universally applicable. When applying this research method to other study areas, urban planners should consider the calculated conversion ratios, local land indicators, and standards, as well as the specific needs and urban planning of the study area. They should conduct comprehensive assessments and site selection for agricultural or constructed land based on the corresponding optimal conversion ratios. Therefore, in future studies, it is necessary to compare the results of this study in southern China with the results of applying this method in other parts of the world to find interesting differences in land use adjustment strategies and comprehensive benefit trade-off dimensions in different countries and regions and obtain necessary references for the implementation of local policies and measures related to ecological corridor construction from these differences.

Additionally, the numerical results obtained from the "Comprehensive Ecological Corridor Benefit Index" assessment model provide ideal reference values for the optimal conversion ratio of agricultural land or constructed land based on specific ecological conditions, population size, land use, and other factors within the planning scope of the ecological corridors. However, the reality is often more complicated and dynamic, and there may be situations that mathematical models fail to capture. Although we provide precise land conversion ratios through the model, it may exhibit overconfidence in practical applications. Therefore, it is necessary for planners to use these ratios as a benchmark and make adjustments based on their expertise and practical experience in optimizing the land structure of ecological corridors. This will ensure that the values are tailored to the specific conditions of the study area. Moreover, the result we presented was an "optimal conversion ratio" for agricultural land or construction land. That is, we did not get the optimal conversion ratio for every land-use subdivision. To do this, we need to set up finer indexes and obtain more detailed data to distinguish the differences

between different subdivisions of land use on the overall benefits of ecological corridor construction, which brings difficulties for us. Additionally, this will undoubtedly make the overall model more complex and cumbersome, and considering that different countries and regions have different standards for land-use subdivisions, it may also bring about limited model promotion. However, exploring the contribution or impact of different land-use subdivisions on the comprehensive benefits of ecological corridor construction can become a further research direction in the future.

## 5. Conclusions

As an integral component of landscape ecological networks, ecological corridors typically encompass a significant portion of existing hybrid landscape land uses in urban areas, including ecological lands, agricultural lands, and constructed lands. From an ecological perspective, maximizing the conversion of non-ecological lands within ecological corridors into ecological lands will yield the maximum ecological benefits and greater provision of ecosystem services. However, in highly competitive metropolitan areas characterized by intense competition for funds and land, the role of landscape planners extends beyond identifying the elements, layout, scale, and corresponding conservation measures of landscape ecological networks [49]. It goes beyond simply achieving improved ecological benefits. A region's landscape ecological network should serve as a comprehensive tool for sustainable development, utilizing knowledge and approaches from multiple dimensions such as ecology, society, economy, and politics to address the region's ecological challenges [49].

In this context, we propose a new indicator called the "Comprehensive Ecological Corridor Benefit Index". This index assesses the comprehensive benefits of ecological, social, and economic aspects achieved through adjusting existing land use and constructing ecological corridors within urban built environments. By maximizing this comprehensive benefit, we can determine the optimal conversion ratio for agricultural land and constructed land within the planning scope of ecological corridors to be converted into ecological land. This will guide the practice of optimizing hybrid land use and landscape structure within the planning scope of ecological corridors in metropolitan areas.

Taking the rapidly urbanizing Su-Jia-Hu area in China as a typical example, we conducted calculations using the proposed index model to determine the optimal conversion ratios of existing agricultural land and constructed land within each of the identified 66 ecological corridors and their respective planning scopes. These calculations aimed to determine the optimal ratios of areas within each ecological corridor that should be converted into ecological land. Based on the optimal conversion ratios for agricultural and constructed lands, we categorized the ecological corridors into three conversion levels: low, medium, and high. Considering the characteristics such as quantity, area, distribution, and location of agricultural and constructed lands within each corridor, we proposed targeted strategies for landscape structure optimization and land use adjustment that are tailored to ecological corridors with varying conversion levels of optimal conversion ratios for agricultural and constructed lands. The index and methodology proposed in this study contribute to the comprehensive consideration and balance of interests among various stakeholders in urban areas and facilitate the implementation of ecological corridor planning outcomes at large spatial and temporal scales.

**Author Contributions:** Conceptualization, J.S.; methodology, J.S.; software, J.S.; validation, J.S. and Y.W.; formal analysis, J.S.; investigation, J.S.; resources, Y.W.; data curation, J.S.; writing—original draft preparation, J.S.; writing—review and editing, Y.W.; visualization, J.S.; supervision, Y.W.; project administration, Y.W.; funding acquisition, J.S. and Y.W. All authors have read and agreed to the published version of the manuscript.

**Funding:** This work was funded by the Key Project of the National Natural Science Foundation of China [grant number 52238003], the National Natural Science Foundation of China [grant number 52208076], and the Fellowship of China Postdoctoral Science Foundation [grant number 2022M712407].

**Data Availability Statement:** Data are contained within the article, and the details are shown in Table 1.

**Conflicts of Interest:** The authors declare no conflict of interest. The funders had no role in the design of the study, in the collection, analyses, or interpretation of the data, in the writing of the manuscript, or in the decision to publish the results.

## Appendix A

Spatial mapping of three key ecosystem services demand for constructed lands in Su-Jia-Hu area.

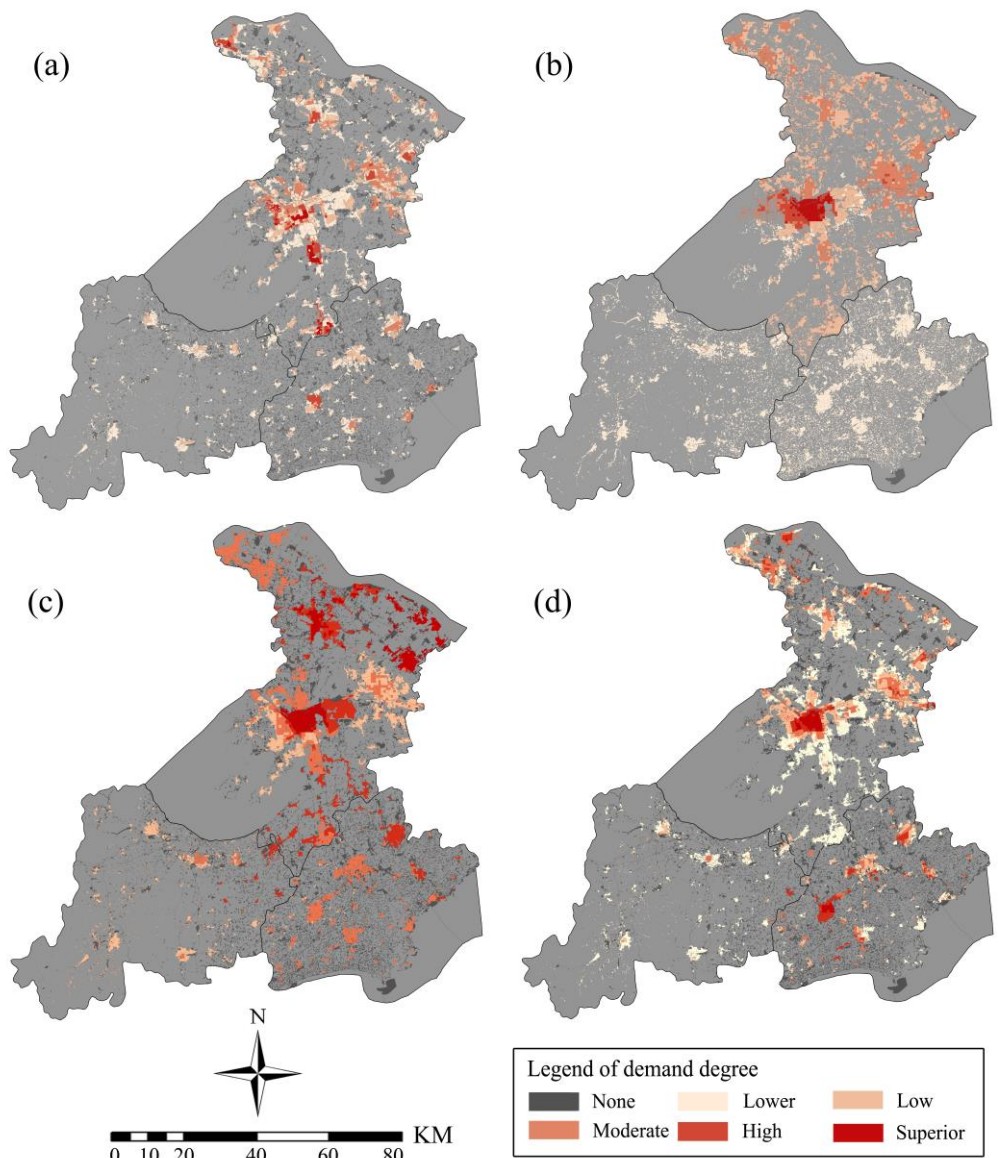

**Figure A1.** Spatial mapping of three key ESs demand of constructed lands in Su-Jia-Hu Arca: (**a**) Flood regulation service; (**b**) Local climate regulation service provided by forest and grassland; (**c**) Local climate regulation service provided by water bodies; (**d**) Outdoor recreation service. (source: [46]).

## Appendix B

Spatial mapping of three key ecosystem services' value of agricultural lands in the Su-Jia-Hu area.

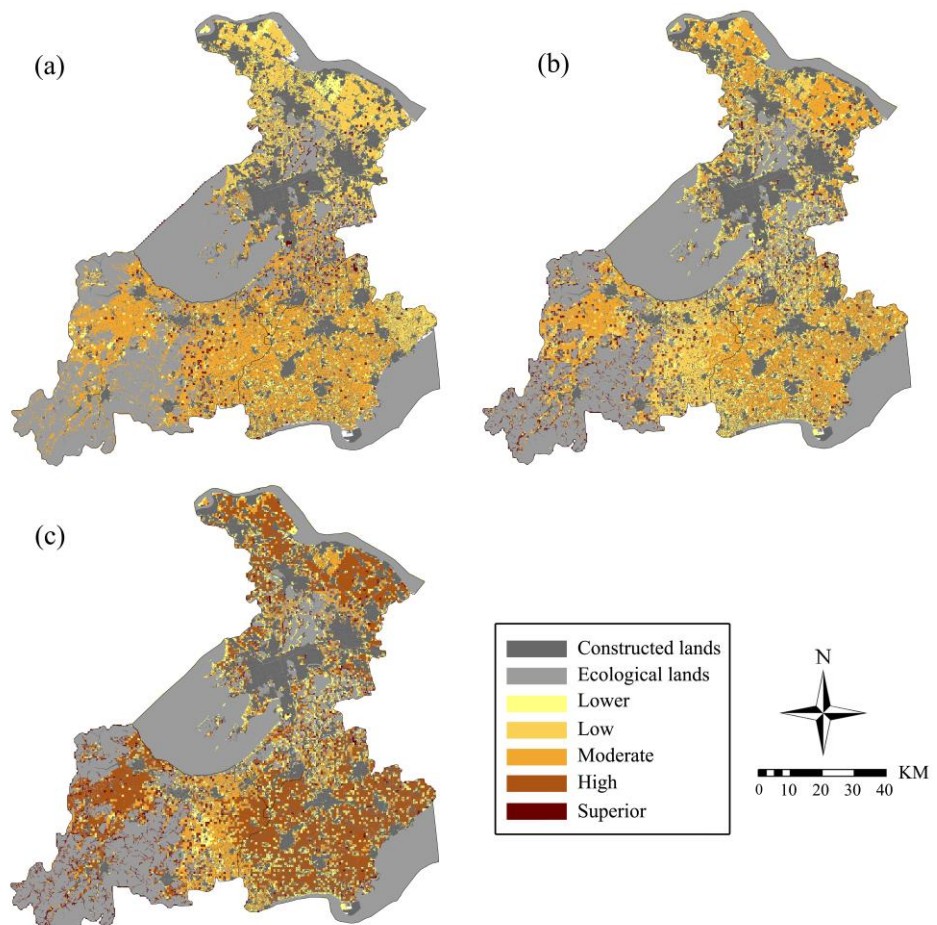

**Figure A2.** Spatial mapping of three key ecosystem services value of agricultural lands in Su-Jia-Hu Area: (**a**) Flood regulation service; (**b**) Local climate regulation service; (**c**) Outdoor recreation service. (source: [48]).

## Appendix C

**Table A1.** Parameter values of 66 ecological corridors used to evaluate their comprehensive benefit index.

| No | Corridor $i$ | $a$ | $b$ | $c$ | $d$ | $m$ | $n$ | $x$ (Maximum of $A$) | $y$ (Maximum of $C$) |
|---|---|---|---|---|---|---|---|---|---|
| 1 | 2 | 12,083.30 | 24,579.77 | 25.00 | 58,521,876.90 | 19,890,396.33 | 1,209,045.78 | 15.65 | 43.47 |
| 2 | 5 | 11,859.27 | 24,350.84 | 25.00 | 32,181,054.40 | 30,152,431.84 | 4,053,077.10 | 22.16 | 78.83 |
| 3 | 12 | 12,040.30 | 24,535.99 | 25.00 | 111,933,675.93 | 123,996,475.20 | 8,204,468.30 | 99.21 | 337.03 |
| 4 | 15 | 11,415.57 | 23,890.97 | 25.00 | 5,447,597.27 | 1,965,288.21 | 164,759.85 | 9.80 | 8.97 |
| 5 | 16 | 11,727.36 | 24,215.04 | 25.00 | 36,090,400.69 | 21,862,238.34 | 1,208,944.97 | 24.01 | 24.84 |
| 6 | 23 | 11,428.07 | 23,904.05 | 25.00 | 24,357,348.46 | 10,133,350.88 | 546,616.69 | 56.42 | 37.51 |
| 7 | 26 | 12,142.87 | 24,640.29 | 25.00 | 73,796,297.00 | 47,039,846.91 | 2,185,146.79 | 13.55 | 15.24 |
| 8 | 29 | 12,031.18 | 24,526.70 | 25.00 | 24,022,871.67 | 18,304,149.99 | 1,533,025.80 | 33.75 | 11.58 |
| 9 | 37 | 11,935.61 | 24,429.09 | 25.00 | 17,918,119.92 | 8,607,462.40 | 326,234.87 | 13.27 | 26.42 |
| 10 | 38 | 12,311.24 | 24,810.52 | 25.00 | 407,930.06 | 196,794.41 | 21,340.42 | 0.31 | 2.38 |

**Table A1.** *Cont.*

| No | Corridor *i* | *a* | *b* | *c* | *d* | *m* | *n* | *x* (Maximum of *A*) | *y* (Maximum of *C*) |
|---|---|---|---|---|---|---|---|---|---|
| 11 | 43 | 11,558.91 | 24,040.50 | 25.00 | 23,872,207.53 | 14,413,167.42 | 492,954.22 | 2.26 | 4.62 |
| 12 | 45 | 11,752.82 | 24,241.31 | 25.00 | 26,872,102.27 | 16,444,894.86 | 650,417.91 | 2.79 | 4.54 |
| 13 | 48 | 8799.53 | 20,975.62 | 25.00 | 12,761,568.10 | 5,731,350.67 | 213,266.53 | 0.19 | 0.32 |
| 14 | 50 | 11,976.22 | 24,470.61 | 25.00 | 104,306,242.00 | 85,461,270.77 | 11,273,686.34 | 176.25 | 970.79 |
| 15 | 58 | 11,457.43 | 23,934.73 | 25.00 | 32,059,899.46 | 17122727.32 | 1,639,091.42 | 5.07 | 20.88 |
| 16 | 62 | 11,301.65 | 23,771.46 | 25.00 | 82,100,430.56 | 43,335,500.05 | 1,067,292.99 | 18.02 | 21.85 |
| 17 | 67 | 12,042.39 | 24,538.12 | 25.00 | 66,649,876.52 | 140,538,476.59 | 15,158,965.97 | 64.84 | 202.51 |
| 18 | 68 | 12,127.26 | 24,624.44 | 25.00 | 88,396,649.79 | 16,1543,896.80 | 19,619,927.42 | 40.96 | 157.16 |
| 19 | 70 | 12,318.25 | 24,817.58 | 25.00 | 105,011,002.80 | 244,688,303.36 | 33,631,915.51 | 30.26 | 72.73 |
| 20 | 72 | 12,155.60 | 24,653.20 | 25.00 | 123,916,499.95 | 166,206,902.34 | 20,852,668.12 | 42.84 | 138.49 |
| 21 | 73 | 11,690.03 | 24,176.47 | 25.00 | 18,454,300.78 | 8,865,096.06 | 258,232.48 | 11.48 | 45.77 |
| 22 | 75 | 12,133.16 | 24,630.42 | 25.00 | 20,042,961.41 | 13,625,451.61 | 893,189.53 | 12.87 | 32.93 |
| 23 | 76 | 11,094.51 | 23,552.61 | 25.00 | 13,444,779.35 | 10,779,199.19 | 1,231,411.90 | 1.99 | 5.39 |
| 24 | 78 | 11,512.50 | 23,992.18 | 25.00 | 9,359,131.81 | 4,347,870.07 | 131,499.42 | 7.75 | 6.78 |
| 25 | 79 | 12,104.10 | 24,600.92 | 25.00 | 15,163,421.12 | 9,594,173.63 | 643,355.31 | 10.74 | 39.85 |
| 26 | 82 | 11,964.73 | 24,458.88 | 25.00 | 15,359,126.26 | 12,882,585.28 | 330,364.25 | 12.05 | 28.26 |
| 27 | 87 | 11,973.86 | 24,468.21 | 25.00 | 41,777,091.95 | 12,684,511.23 | 161,910.56 | 17.89 | 12.16 |
| 28 | 88 | 11,994.08 | 24,488.86 | 25.00 | 77,123,255.77 | 31,388,541.31 | 2,157,249.04 | 35.73 | 146.64 |
| 29 | 103 | 12,015.19 | 24,510.40 | 25.00 | 21,718,149.19 | 11,745,113.36 | 312,811.53 | 32.74 | 62.38 |
| 30 | 104 | 11,713.71 | 24,200.94 | 25.00 | 8,963,188.02 | 32,892,833.99 | 4,917,848.03 | 0.15 | 0.04 |
| 31 | 107 | 12,036.53 | 24,532.16 | 25.00 | 67,271,598.31 | 79,443,575.37 | 10,093,084.06 | 25.23 | 70.79 |
| 32 | 108 | 11,982.06 | 24,476.58 | 25.00 | 102,616,993.18 | 167,611,091.09 | 22,071,160.88 | 63.95 | 197.31 |
| 33 | 200 | 12,107.50 | 24,604.37 | 25.00 | 65,281,711.28 | 26,644,639.52 | 2,675,012.74 | 44.82 | 542.40 |
| 34 | 201 | 11,977.67 | 24,472.10 | 25.00 | 190,098,347.78 | 42,066,581.64 | 1,450,453.95 | 31.48 | 347.86 |
| 35 | 202 | 10,683.96 | 23,112.72 | 25.00 | 93,889,156.42 | 53,843,664.72 | 6,506,430.79 | 15.12 | 39.61 |
| 36 | 203 | 11,820.23 | 24,310.73 | 25.00 | 79,869,239.73 | 38,374,486.57 | 4,593,803.14 | 14.74 | 224.27 |
| 37 | 204 | 11,715.92 | 24,203.22 | 25.00 | 132,544,605.70 | 150,310,498.54 | 16,566,577.18 | 8.92 | 86.70 |
| 38 | 205 | 12,100.36 | 24,597.11 | 25.00 | 44,315,460.59 | 64,281,905.16 | 18,422,053.32 | 43.72 | 203.45 |
| 39 | 206 | 5131.00 | 16,017.18 | 25.00 | 27,0662,939.47 | 6,739,5851.40 | 4,153,113.86 | 5.63 | 21.06 |
| 40 | 207 | 12,118.75 | 24,615.79 | 25.00 | 96,280,044.07 | 107,625,167.55 | 16,949,153.91 | 48.30 | 216.11 |
| 41 | 208 | 11,804.20 | 24,294.23 | 25.00 | 269,950,225.63 | 119,440,111.97 | 2,787,853.15 | 115.65 | 306.66 |
| 42 | 210 | 11,447.46 | 23,924.32 | 25.00 | 104,065,841.43 | 36,829,042.20 | 2,340,925.31 | 16.49 | 47.43 |
| 43 | 212 | 11,961.64 | 24,455.72 | 25.00 | 106,369,623.21 | 42,245,577.97 | 1,018,791.88 | 42.61 | 94.97 |
| 44 | 215 | 11,991.77 | 24,486.50 | 25.00 | 66,700,810.22 | 24,018,749.41 | 933,111.81 | 24.81 | 59.16 |
| 45 | 216 | 12,085.19 | 24,581.69 | 25.00 | 362,677,887.22 | 138,785,829.64 | 5,021,232.05 | 46.89 | 82.42 |
| 46 | 217 | 11,784.99 | 24,274.46 | 25.00 | 94,860,489.83 | 17,154,689.59 | 701,815.80 | 8.90 | 49.78 |
| 47 | 218 | 11,958.21 | 24,452.21 | 25.00 | 37,837,370.54 | 30,026,022.74 | 4,136,522.46 | 17.15 | 19.85 |
| 48 | 221 | 10,512.87 | 22,926.92 | 25.00 | 207,156,730.48 | 81,348,724.36 | 7,651,819.08 | 4.67 | 42.02 |
| 49 | 223 | 12,330.41 | 24,829.83 | 25.00 | 183,506,191.13 | 131,455,320.56 | 9,108,699.74 | 10.01 | 33.86 |
| 50 | 228 | 11,425.65 | 23,901.51 | 25.00 | 69,949,148.24 | 74,046,092.55 | 17,641,543.05 | 2.14 | 49.97 |
| 51 | 229 | 11,908.11 | 24,400.94 | 25.00 | 31,379,859.47 | 9,101,412.95 | 625,441.94 | 12.35 | 72.08 |
| 52 | 230 | 12,020.01 | 24,515.31 | 25.00 | 169,127,576.69 | 109,130,880.26 | 3,450,642.37 | 32.95 | 68.72 |
| 53 | 231 | 11,620.40 | 24,104.36 | 25.00 | 67,646,135.00 | 51,610,074.71 | 1,916,992.26 | 19.21 | 24.55 |
| 54 | 301 | 11,901.77 | 24,394.44 | 25.00 | 256,918,118.87 | 163,469,058.38 | 24,856,516.12 | 4.27 | 63.45 |
| 55 | 302 | 11,945.39 | 24,439.10 | 25.00 | 45,985,400.39 | 32174097.58 | 5,627,706.27 | 35.70 | 378.21 |
| 56 | 303 | 0.00 | 0.00 | 25.00 | 972,415,710.79 | 376,235,897.35 | 34,496,751.03 | 1.64 | 6.74 |

**Table A1.** *Cont.*

| No | Corridor *i* | *a* | *b* | *c* | *d* | *m* | *n* | *x* (Maximum of *A*) | *y* (Maximum of *C*) |
|----|----|----|----|----|----|----|----|----|----|
| 57 | 304 | 1013.49 | 7118.61 | 25.00 | 535,466,948.17 | 537,093,162.81 | 7,738,110.66 | 0.79 | 42.99 |
| 58 | 305 | 8929.37 | 21,129.80 | 25.00 | 97,888,691.14 | 40,405,188.77 | 4,046,026.67 | 1.56 | 13.03 |
| 59 | 306 | 12,221.03 | 24,719.46 | 25.00 | 415,940,724.96 | 208,860,920.37 | 4,663,956.29 | 10.47 | 13.31 |
| 60 | 307 | 10,905.86 | 23,351.51 | 25.00 | 342,005,289.96 | 133,897,528.56 | 13,097,126.64 | 2.24 | 20.24 |
| 61 | 308 | 12,317.71 | 24,817.04 | 25.00 | 178,951,800.79 | 161,445,509.44 | 4,576,161.02 | 7.70 | 26.96 |
| 62 | 309 | 12,146.86 | 24,644.33 | 25.00 | 157,193,469.10 | 142,670,827.10 | 1,607,304.42 | 18.33 | 51.32 |
| 63 | 310 | 12,077.60 | 24,573.97 | 25.00 | 69,936,085.33 | 28,438,522.86 | 1,563,523.28 | 10.89 | 23.56 |
| 64 | 311 | 9518.03 | 21,815.17 | 25.00 | 210,591,209.52 | 65,949,168.25 | 5,054,707.48 | 1.00 | 10.62 |
| 65 | 312 | 12,062.54 | 24,558.64 | 25.00 | 100,328,996.67 | 45,649,786.64 | 1,224,148.69 | 36.45 | 35.21 |
| 66 | 313 | 10,683.96 | 23,112.72 | 25.00 | 93,889,156.42 | 53,843,664.72 | 6,506,430.79 | 15.12 | 39.61 |

## Appendix D

**Table A2.** Statistics of the area, proportion, and optimal conversion ratio of agricultural land in 66 ecological corridors.

| No | Corridor *i* | Planned Area of Corridor (km²) | Total Area of Agricultural Land in Corridor (km²) | The Proportion of Agricultural Land in the Corridor | Conversion Area of Agricultural Land (km²) | Optimal Conversion Ratio of Agricultural Land | Conversion Degree |
|----|----|----|----|----|----|----|----|
| 1 | 2 | 113.15 | 15.65 | 14% | 7.31 | 47% | Moderate |
| 2 | 5 | 69.64 | 22.16 | 32% | 11.96 | 54% | Moderate |
| 3 | 12 | 185.23 | 99.21 | 54% | 7.91 | 8% | Low |
| 4 | 15 | 64.03 | 9.80 | 15% | 9.80 | 100% | High |
| 5 | 16 | 98.11 | 24.01 | 24% | 10.97 | 46% | Moderate |
| 6 | 23 | 234.11 | 56.42 | 24% | 10.94 | 19% | Low |
| 7 | 26 | 17.21 | 13.55 | 79% | 10.48 | 77% | High |
| 8 | 29 | 61.45 | 33.75 | 55% | 16.86 | 50% | Moderate |
| 9 | 37 | 27.22 | 13.27 | 49% | 13.27 | 100% | High |
| 10 | 38 | 1.67 | 0.31 | 19% | 0.31 | 100% | High |
| 11 | 43 | 3.76 | 2.26 | 60% | 2.26 | 100% | High |
| 12 | 45 | 5.32 | 2.79 | 52% | 2.79 | 100% | High |
| 13 | 48 | 0.32 | 0.19 | 60% | 0.19 | 100% | High |
| 14 | 50 | 394.54 | 176.25 | 45% | 8.58 | 5% | Low |
| 15 | 58 | 9.68 | 5.07 | 52% | 5.07 | 100% | High |
| 16 | 62 | 94.21 | 18.02 | 19% | 8.83 | 49% | Moderate |
| 17 | 67 | 118.64 | 64.84 | 55% | 18.79 | 29% | Low |
| 18 | 68 | 57.53 | 40.96 | 71% | 13.51 | 33% | Low |
| 19 | 70 | 17.45 | 6.05 | 35% | 3.33 | 55% | Moderate |
| 20 | 72 | 64.75 | 42.84 | 66% | 9.62 | 22% | Low |
| 21 | 73 | 55.30 | 11.48 | 21% | 10.56 | 92% | High |
| 22 | 75 | 60.69 | 12.87 | 21% | 11.49 | 89% | High |
| 23 | 76 | 3.02 | 1.99 | 66% | 1.99 | 100% | High |
| 24 | 78 | 14.56 | 7.75 | 53% | 7.75 | 100% | High |
| 25 | 79 | 34.61 | 10.74 | 31% | 10.74 | 100% | High |

**Table A2.** *Cont.*

| No | Corridor *i* | Planned Area of Corridor (km²) | Total Area of Agricultural Land in Corridor (km²) | The Proportion of Agricultural Land in the Corridor | Conversion Area of Agricultural Land (km²) | Optimal Conversion Ratio of Agricultural Land | Conversion Degree |
|---|---|---|---|---|---|---|---|
| 26 | 82 | 45.63 | 12.05 | 26% | 12.05 | 100% | High |
| 27 | 87 | 24.72 | 17.89 | 72% | 14.55 | 81% | High |
| 28 | 88 | 96.16 | 35.73 | 37% | 7.15 | 20% | Low |
| 29 | 103 | 62.31 | 32.74 | 53% | 8.29 | 25% | Low |
| 30 | 104 | 0.18 | 0.15 | 82% | 0.15 | 102% | High |
| 31 | 107 | 36.03 | 25.23 | 70% | 10.60 | 42% | Moderate |
| 32 | 108 | 121.09 | 63.95 | 53% | 11.56 | 18% | Low |
| 33 | 200 | 270.68 | 44.82 | 17% | 8.06 | 18% | Low |
| 34 | 201 | 153.54 | 31.48 | 21% | 4.62 | 15% | Low |
| 35 | 202 | 16.86 | 15.12 | 90% | 8.11 | 54% | Moderate |
| 36 | 203 | 39.60 | 14.74 | 37% | 8.14 | 55% | Moderate |
| 37 | 204 | 10.04 | 8.92 | 89% | 8.54 | 96% | High |
| 38 | 205 | 68.29 | 43.72 | 64% | 15.19 | 35% | Low |
| 39 | 206 | 9.39 | 5.63 | 60% | 5.63 | 100% | High |
| 40 | 207 | 29.43 | 12.94 | 44% | 2.66 | 21% | Low |
| 41 | 208 | 226.14 | 115.65 | 51% | 4.18 | 4% | Low |
| 42 | 210 | 58.14 | 16.49 | 28% | 6.44 | 39% | Moderate |
| 43 | 212 | 86.48 | 42.61 | 49% | 5.24 | 12% | Low |
| 44 | 215 | 37.48 | 24.81 | 66% | 6.47 | 26% | Low |
| 45 | 216 | 95.22 | 46.89 | 49% | 4.20 | 9% | Low |
| 46 | 217 | 13.32 | 8.90 | 67% | 6.30 | 71% | Moderate |
| 47 | 218 | 16.23 | 15.56 | 96% | 10.46 | 67% | Moderate |
| 48 | 221 | 6.91 | 4.67 | 68% | 4.67 | 100% | High |
| 49 | 223 | 2.17 | 0.48 | 22% | 0.30 | 63% | Moderate |
| 50 | 228 | 8.47 | 2.14 | 25% | 2.14 | 100% | High |
| 51 | 229 | 15.14 | 12.35 | 82% | 8.60 | 70% | Moderate |
| 52 | 230 | 27.52 | 17.50 | 64% | 2.81 | 16% | Low |
| 53 | 231 | 32.61 | 19.21 | 59% | 8.30 | 43% | Moderate |
| 54 | 301 | 5.44 | 4.27 | 78% | 4.27 | 100% | High |
| 55 | 302 | 67.83 | 35.70 | 53% | 10.78 | 30% | Low |
| 56 | 303 | 4.00 | 1.64 | 41% | 0.00 | 0% | Low |
| 57 | 304 | 2.00 | 0.79 | 39% | 0.79 | 100% | High |
| 58 | 305 | 3.21 | 1.56 | 49% | 1.56 | 100% | High |
| 59 | 306 | 2.38 | 0.86 | 36% | 0.50 | 58% | Moderate |
| 60 | 307 | 1.94 | 0.09 | 5% | 0.09 | 100% | High |
| 61 | 308 | 0.97 | 0.16 | 16% | 0.11 | 71% | Moderate |
| 62 | 309 | 16.30 | 0.41 | 3% | 0.10 | 24% | Low |
| 63 | 310 | 15.76 | 10.89 | 69% | 9.24 | 85% | High |
| 64 | 311 | 1.51 | 1.00 | 66% | 1.00 | 100% | High |
| 65 | 312 | 50.13 | 36.45 | 73% | 6.29 | 17% | Low |
| 66 | 215 | 37.48 | 24.81 | 66% | 6.47 | 26% | Low |

## Appendix E

**Table A3.** Pearson's correlation analysis results of ecological corridor area, agricultural land area, and optimal conversion area and ratio.

| | | Ecological Corridor Area | Total Area of Agricultural Land in Corridor | Optimal Conversion Ratio of Agricultural Land | Optimal Conversion Area of Agricultural Land |
|---|---|---|---|---|---|
| Ecological corridor area | Pearson's correlation | 1 | 0.886 ** | −0.612 ** | 0.325 ** |
| | Significance (two sides) | | 0.000 | 0.000 | 0.008 |
| | N | 66 | 66 | 66 | 66 |
| Total area of agricultural land in corridor | Pearson's correlation | 0.886 ** | 1 | −0.644 ** | 0.362 ** |
| | Significance (two sides) | 0.000 | | 0.000 | 0.003 |
| | N | 66 | 66 | 66 | 66 |
| Optimal conversion ratio of agricultural land | Pearson's correlation | −0.612 ** | −0.644 ** | 1 | −0.181 |
| | Significance (two sides) | 0.000 | 0.000 | | 0.148 |
| | N | 66 | 66 | 66 | 66 |
| Optimal conversion area of agricultural land | Pearson correlation | 0.325 ** | 0.362 ** | −0.181 | 1 |
| | Significance (two sides) | 0.008 | 0.003 | 0.148 | |
| | N | 66 | 66 | 66 | 66 |

** Significantly correlated at the level of 0.01 (two-sided test).

## Appendix F

**Table A4.** Statistics of the area, proportion, and optimal conversion ratio of constructed land in 66 ecological corridors.

| No | Corridor *i* | Planned Area of Corridor (km²) | Total Area of Constructed Land in Corridor(km²) | The Proportion of Constructed Land in the Corridor | Conversion Area of Constructed Land (km²) | Optimal Conversion Ratio of Constructed Land | Conversion Degree |
|---|---|---|---|---|---|---|---|
| 1 | 2 | 113.15 | 6.35 | 6% | 6.35 | 100% | High |
| 2 | 5 | 69.64 | 8.28 | 12% | 2.08 | 25% | Low |
| 3 | 12 | 185.23 | 35.17 | 19% | 1.71 | 5% | Low |
| 4 | 15 | 64.03 | 0.90 | 1% | 0.90 | 100% | High |
| 5 | 16 | 98.11 | 2.49 | 3% | 2.49 | 100% | High |
| 6 | 23 | 234.11 | 3.75 | 2% | 3.75 | 100% | High |
| 7 | 26 | 17.21 | 1.53 | 9% | 1.53 | 100% | High |
| 8 | 29 | 61.45 | 1.16 | 2% | 1.16 | 100% | High |

**Table A4.** *Cont.*

| No | Corridor $i$ | Planned Area of Corridor (km$^2$) | Total Area of Constructed Land in Corridor(km$^2$) | The Proportion of Constructed Land in the Corridor | Conversion Area of Constructed Land (km$^2$) | Optimal Conversion Ratio of Constructed Land | Conversion Degree |
|---|---|---|---|---|---|---|---|
| 9 | 37 | 27.22 | 2.64 | 10% | 2.64 | 100% | High |
| 10 | 38 | 1.67 | 0.24 | 14% | 0.24 | 100% | High |
| 11 | 43 | 3.76 | 0.49 | 13% | 0.49 | 100% | High |
| 12 | 45 | 5.32 | 0.45 | 9% | 0.45 | 100% | High |
| 13 | 48 | 0.32 | 0.03 | 10% | 0.03 | 99% | High |
| 14 | 50 | 394.54 | 100.97 | 26% | 1.72 | 2% | Low |
| 15 | 58 | 9.68 | 2.09 | 22% | 2.09 | 100% | High |
| 16 | 62 | 94.21 | 2.18 | 2% | 2.18 | 100% | High |
| 17 | 67 | 118.64 | 21.34 | 18% | 0.00 | 0% | Low |
| 18 | 68 | 57.53 | 15.78 | 27% | 0.46 | 3% | Low |
| 19 | 70 | 17.45 | 7.49 | 43% | 0.00 | 0% | Low |
| 20 | 72 | 64.75 | 14.11 | 22% | 1.00 | 7% | Low |
| 21 | 73 | 55.30 | 4.66 | 8% | 4.66 | 100% | High |
| 22 | 75 | 60.69 | 3.30 | 5% | 3.30 | 100% | High |
| 23 | 76 | 3.02 | 0.55 | 18% | 0.55 | 100% | High |
| 24 | 78 | 14.56 | 0.82 | 6% | 0.82 | 100% | High |
| 25 | 79 | 34.61 | 4.36 | 13% | 4.36 | 100% | High |
| 26 | 82 | 45.63 | 2.88 | 6% | 2.88 | 100% | High |
| 27 | 87 | 24.72 | 1.22 | 5% | 1.22 | 100% | High |
| 28 | 88 | 96.16 | 14.79 | 15% | 3.57 | 24% | Low |
| 29 | 103 | 62.31 | 6.24 | 10% | 5.69 | 91% | High |
| 30 | 104 | 0.18 | 0.00 | 3% | 0.00 | 96% | High |
| 31 | 107 | 36.03 | 7.58 | 21% | 1.43 | 19% | Low |
| 32 | 108 | 121.09 | 20.37 | 17% | 0.71 | 3% | Low |
| 33 | 200 | 270.68 | 54.71 | 20% | 3.24 | 6% | Low |
| 34 | 201 | 153.54 | 34.77 | 23% | 4.90 | 14% | Low |
| 35 | 202 | 16.86 | 5.05 | 30% | 2.95 | 58% | Moderate |
| 36 | 203 | 39.60 | 25.53 | 64% | 2.99 | 12% | Low |
| 37 | 204 | 10.04 | 8.71 | 87% | 1.28 | 15% | Low |
| 38 | 205 | 68.29 | 20.93 | 31% | 0.74 | 4% | Low |
| 39 | 206 | 9.39 | 2.38 | 25% | 2.38 | 100% | High |
| 40 | 207 | 29.43 | 23.40 | 80% | 1.28 | 5% | Low |
| 41 | 208 | 226.14 | 30.98 | 14% | 3.60 | 12% | Low |
| 42 | 210 | 58.14 | 4.74 | 8% | 3.68 | 77% | Moderate |
| 43 | 212 | 86.48 | 9.52 | 11% | 4.72 | 50% | Moderate |

**Table A4.** *Cont.*

| No | Corridor $i$ | Planned Area of Corridor (km$^2$) | Total Area of Constructed Land in Corridor(km$^2$) | The Proportion of Constructed Land in the Corridor | Conversion Area of Constructed Land (km$^2$) | Optimal Conversion Ratio of Constructed Land | Conversion Degree |
|----|----|----|----|----|----|----|----|
| 44 | 215 | 37.48 | 5.95 | 16% | 4.79 | 81% | Moderate |
| 45 | 216 | 95.22 | 8.25 | 9% | 3.09 | 37% | Low |
| 46 | 217 | 13.32 | 10.51 | 79% | 10.51 | 100% | High |
| 47 | 218 | 16.23 | 2.23 | 14% | 2.23 | 100% | High |
| 48 | 221 | 6.91 | 4.52 | 65% | 3.33 | 74% | Moderate |
| 49 | 223 | 2.17 | 0.88 | 41% | 0.52 | 59% | Moderate |
| 50 | 228 | 8.47 | 4.99 | 59% | 3.60 | 72% | Moderate |
| 51 | 229 | 15.14 | 7.83 | 52% | 5.78 | 74% | Moderate |
| 52 | 230 | 27.52 | 6.88 | 25% | 2.89 | 42% | Moderate |
| 53 | 231 | 32.61 | 2.48 | 8% | 2.48 | 100% | High |
| 54 | 301 | 5.44 | 5.08 | 93% | 1.71 | 34% | Low |
| 55 | 302 | 67.83 | 38.32 | 56% | 2.07 | 5% | Low |
| 56 | 303 | 4.00 | 0.90 | 22% | 0.90 | 100% | High |
| 57 | 304 | 2.00 | 0.84 | 42% | 0.84 | 100% | High |
| 58 | 305 | 3.21 | 1.48 | 46% | 1.48 | 100% | High |
| 59 | 306 | 2.38 | 1.34 | 56% | 1.34 | 100% | High |
| 60 | 307 | 1.94 | 1.72 | 89% | 1.72 | 100% | High |
| 61 | 308 | 0.97 | 0.78 | 80% | 0.67 | 87% | High |
| 62 | 309 | 16.30 | 6.08 | 37% | 3.87 | 64% | Low |
| 63 | 310 | 15.76 | 2.36 | 15% | 2.36 | 100% | High |
| 64 | 311 | 1.51 | 1.13 | 75% | 1.13 | 100% | High |
| 65 | 312 | 50.13 | 3.52 | 7% | 3.52 | 100% | High |
| 66 | 215 | 37.48 | 5.95 | 16% | 4.79 | 81% | Moderate |

## Appendix G

**Table A5.** Pearson's correlation analysis results of the ecological corridor area, constructed land area, and optimal conversion area and ratio.

| | | Ecological Corridor Area | Total Area of Constructed Land in Corridor | Optimal Conversion Ratio Of Constructed Land | Optimal Conversion Area Of Constructed Land |
|----|----|----|----|----|----|
| Ecological corridor area | Pearson's correlation | 1 | 0.813 ** | 0.167 | −0.462 ** |
| | Significance (two sides) | | 0.000 | 0.184 | 0.000 |
| | N | 66 | 66 | 66 | 66 |

**Table A5.** *Cont.*

| | | Ecological Corridor Area | Total Area of Constructed Land in Corridor | Optimal Conversion Ratio Of Constructed Land | Optimal Conversion Area Of Constructed Land |
|---|---|---|---|---|---|
| Total area of constructed land in corridor | Pearson's correlation | 0.813 ** | 1 | 0.068 | −0.668 ** |
| | Significance (two sides) | 0.000 | | 0.592 | 0.000 |
| | N | 66 | 66 | 66 | 66 |
| Optimal conversion ratio of constructed land | Pearson's correlation | 0.167 | 0.068 | 1 | 0.125 |
| | Significance (two sides) | 0.184 | 0.592 | | 0.322 |
| | N | 66 | 66 | 66 | 66 |
| Optimal conversion area of constructed land | Pearson's correlation | −0.462 ** | −0.668 ** | 0.125 | 1 |
| | Significance (two sides) | 0.000 | 0.000 | 0.322 | |
| | N | 66 | 66 | 66 | 66 |

** Significantly correlated at the level of 0.01 (two-sided test).

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
