# Peer review of "Optimizing Landscape Structure of Hybrid Land Use in Ecological Corridors Based on Comprehensive Benefit Index in Metropolitan Area"

_forests, doi:10.3390/f14091714_

Round 1
Reviewer 1 Report
This manuscript developing a benefit index for ecological corridors assessment provides an interesting case study for the Su-Jia-Hu area, China. However, the manuscript is in some parts difficult to follow because of the following:
The Authors seem to give for granted that the process associated with ecological conversion or optimization is well known by the readership. This might be so, but for novice readers it become difficult to follow the manuscript. The Authors should clarify the basic concepts of ecological land as opposed to natural land, for those not familiar with that; likewise for ecological environment and ecological resistence concepts. To this end it is suggested to develop a literature review on the background of these concepts and highlight differences with concepts which might look similar like ecology of transportation infrastructure and road ecology, and the likes.
Methodology: Too concise. Please, fully explain figure 1 to enable replicability of the proposed methodology elsewhere.
Equations presented: Some examples of calculation would help understand them and clarify the meaning of parameters like k1, which are swiftly explained. However, it is ok to have an additional explanation in the appendix, but the main text must be self-explicatory even without the appendix.
On this, continuously referring to the appendices compels the reader to go back and forth between the main text and each of them, which is a little bit annoying and slackens the reading pace. Please, amend this especially in the result parts.
Section on results: To better understand results and especially the pictures, maybe satellite images should be added.
Is there any regulatory framework to support this analysis?
The land use description is rather general: What types of built and natural environments are we dealing with? What is the relevance/impact of transportation infrastructures in the indexes provided?
Eventually, a language check is strongly suggested. Typos to amend (for example, fig. 2 "lengend".
See last point above
Author Response
Response to the reviewers’ comments
Reviewer 1:
- The Authors seem to give for granted that the process associated with ecological conversion or optimization is well known by the readership. This might be so, but for novice readers it become difficult to follow the manuscript. The Authors should clarify the basic concepts of ecological land as opposed to natural land, for those not familiar with that; likewise for ecological environment and ecological resistance concepts. To this end it is suggested to develop a literature review on the background of these concepts and highlight differences with concepts which might look similar like ecology of transportation infrastructure and road ecology, and the likes.
- Response:
Thank you very much to the reviewer for pointing out the above issues for us. We think your suggestions are necessary to help our manuscript be understood and follow by a wider audience, especially novice readers. Therefore, first of all, we have supplemented and modified the Introduction section, mainly including: 1) Adding a literature review on the concept, role, and importance of ecological land, emphasizing its role in metropolitan area settings, and providing corresponding references, thus distinguishing it from the concept of "natural land"; 2) Complementing the concept of "ecological environment" and further explains its relationship with ecological land; 3) Complementing the concept of "ecological corridors" and their role in metropolitan areas, and clarified the differences between them and similar concepts such as transportation infrastructure, ecological roads, etc., which also provide connectivity in cities. The above three changes are as follows:
1) “Ecological land is the land with important ecological functions and provides eco-system services essential for the maintenance of regional ecological security [1], which not only supports and protects the ecosystem stability, the virtuous cycle and sustain-able development [2], but also provides the space necessary for basic ecological activities for the inhabitants of metropolitan areas [3]. Although there is currently no uniform definition of ecological land, it has been identified as a distinct land type in many studies [4-6], especially in metropolitan areas.”
2) “The ecological environment is a complex of natural factors that have an impact on human survival and development [8]. As the important component and constitution, the degradation and loss of ecological lands seriously threaten the stability of the eco-logical environment and the well-being of the human beings living in it.”
3) “Ecological corridors are strip-like or linear ecological landscapes with a certain spatial range [14,15], which are heterogeneous compared with the surrounding landscape [16]. They act as bridges and connections between various ecological source areas, and are also crucial for achieving urban sustainable development [17]. Unlike the linear transportation infrastructure and ecological roads in the city, which also play a role in connecting between destinations and deliver goods and materials, ecological corridors also play the functions of natural habitat, green open space, hu-man habitat isolation , and human settlement environment beautification and enhancement [16,18], which are an inherent part of the landscape ecological network [16] and one of the concrete embodiments of the ecological environment in the metropolitan areas.”
In addition, we have supplemented the concept of "ecological resistance" in subsection 2.1 of the Methodology section (where it first appears) and provided corresponding references. The specific modifications are as follows:
“(2) “Economic input cost (R)", which represents the economic costs associated with the construction of the ecological corridor. The conversion between different types of land use is a process of mutual conversion or competition for control and coverage of space, which is achieved by overcoming various resistances [44]. In the process of converting non-ecological land to ecological land, it is necessary to overcome the "ecological resistance" caused by ecological factors such as topography, slope, vegetation cover, and land cover [45,46]. The process of overcoming these ecological resistances means that construction costs are commensurate.”
- Methodology: Too concise. Please, fully explain figure 1 to enable replicability of the proposed methodology elsewhere.
- Response:
Thank you very much for the comments and suggestions of the reviewer. Based on your suggestions, we have supplemented the interpretation of Figure 1 in Section 2.1. It mainly included the following aspects: 1) supplementing the connotation and role of E, R and S factors; 2) the definition of "ecological resistance" and its influencing factors were supplemented; 3) The functional relationship and calculation method of agricultural land area A and the constructed land area C converted into ecological land in the ecological corridor and E, R and S were further explained. 4) pointing out the meaning and role of using A and C to characterize I, that is, the complex problem of land use conversion and comprehensive benefit calculation was simplified to: " how much cultivated land/ constructed land needs to be converted into ecological land in an ecological corridor to maximize the comprehensive benefits of the corridor?" In addition, we have mapped the color of the block diagram in Figure 1 to the text narrative to help the reader keep up with the interpretation of Figure 1 at every step. The specific changes are as follows:
1) “This indicator is a function of three factors (Figure 1- orange box): (1) "Ecological bene-fit enhancement potential (E)", which represents the ecological benefits achieved through ecological corridor construction. When existing cultivated land or constructed land is converted to ecological land, its capacity to provide multiple ecosystem services will be enhanced, bringing additional ecological benefits to the ecological corridors. (2) “Economic input cost (R)", which represents the economic costs associated with the construction of the ecological corridor. The conversion between different types of land use is a process of mutual conversion or competition for control and coverage of space, which is achieved by overcoming various resistances [44]. In the process of converting non-ecological land to ecological land, it is necessary to overcome the "ecological resistance" caused by ecological factors such as topography, slope, vegetation cover, and land cover [45,46]. The process of overcoming these ecological resistances means that construction costs are commensurate. (3) “Social coordination cost (S)", which represents the social costs involved in the construction of the ecological corridor. When non-ecological land within an ecological corridor is converted to ecological land, indigenous peoples and stakeholders are implicated [38], requiring the government to compensate or evict it at a cost [39]. Therefore, the total population in non-ecological lands within ecological corridors that need to be converted into ecological lands can reflect the social coordination costs in the process of building ecological corridors through land use adjustment.”
2) “The conversion between different types of land use is a process of mutual conversion or competition for control and coverage of space, which is achieved by overcoming various resistances [44]. In the process of converting non-ecological land to ecological land, it is necessary to overcome the "ecological resistance" caused by ecological factors such as topography, slope, vegetation cover, and land cover [45,46].”
3) “We calculated E by multiplying the average improvement of multiple ecosystem ser-vices per unit area of cultivated land (constructed land) per unit area in the ecological corridor and A (C). R was calculated by the product of the modified ecological resistance value of cultivated land (constructed land) per unit area in the ecological corridor and A(C). S was calculated by the product of population density of the study area and A(C). This step established the functional relationship between the agricultural land area (A) and constructed land area (C) converted into ecological lands within the ecological corridor, and the "Comprehensive Ecological Corridor Benefit Index (I)."”
4) “This area ratio can provide a reference for the quantifiable threshold for the conversion of non-ecological land to ecological land within the ecological corridor, thereby simplifying the above complex question as: how much cultivated land/ constructed land needs to be converted into ecological land in an ecological corridor to maximize the comprehensive benefits of the corridor?”
- Equations presented: Some examples of calculation would help understand them and clarify the meaning of parameters like k1, which are swiftly explained. However, it is ok to have an additional explanation in the appendix, but the main text must be self-explicatory even without the appendix.
- Response:
Thank you very much to the reviewer for your suggestions above. Based on your comments, first of all, we have added clarification of the meaning of constant , specific values set for them, and the reason for set. The specific modifications are as follows:
“ are constants, and they are set so that the impact of E, R, and S on I is of the same magnitude, because here we assumed that the impact of ecological, social, and economic aspects on the comprehensive benefits of ecological corridors is equivalent. Considering the value ranges of E, R, and S, we set =10000000, =0.0000000001, and =0.1.”
Second, we have moved the table "Parameter values of 66 ecological corridors used to evaluate their comprehensive benefit index" originally located in Supplementary Material to the Appendix at the end of the main text as the new Appendix C. This table provides the specific values of parameters a, b, d, e, m, and n obtained by substituting the relevant calculation data into Formula (4) and expanding it, which can provide readers with a reference for related methods more conveniently.
- On this, continuously referring to the appendices compels the reader to go back and forth between the main text and each of them, which is a little bit annoying and slackens the reading pace. Please, amend this especially in the result parts.
- Response:
Thank you to the reviewer for pointing out the above shortcomings for us. Based on your suggestions, we have made the following modifications:
1) We moved the original Appendices A and B to the main text as Figure 3 and Table 2. Because these two charts show the basic data of this study, namely "Spatial distribution of 66 ecological corridors and their widths in Su-Jia-Hu area" and "The suitable construction width ranges of 66 ecological corridors and detailed corridor information". Based on the reviewer's suggestions, we thought they could be placed directly in the text to visually present the 66 ecological corridors and their width information for the readers.
2) For the Appendices to the results section, "Statistics of the area, proportion, and optimal conversion ratio of agricultural land in 66 ecological corridors" and "Statistics of the area, proportion, and optimal conversion ratio of constructed land in 66 ecological corridors" are long and show computational process data, so we have retained their place in the Appendix. We moved the original Appendix I, "Pearson correlation analysis results of optimal conversion ratio of agricultural land and constructed land", to the main text as a new Table 5, making it representative of the results of our SPSS correlation analysis in the main text.
3) We moved the table "Parameter values of 66 ecological corridors used to evaluate their comprehensive benefit index" originally located in Supplementary Material to the appendix at the end of the main text as the new Appendix C, to provide readers with a more convenient reference.
- Section on results: To better understand results and especially the pictures, maybe satellite images should be added.
- Response:
Thank you very much for your comments. We also strongly agree with the reviewer's suggestion that satellite imagery provide a more realistic and clearer picture of the current status and conversion of land use within the planning scope of ecological corridor. However, since the distribution of 66 ecological corridors in the study area and their planning scope, which are the basic data of this study, are not planned corridors that already exist in reality, but potential ecological corridors that have been delineated by corresponding methods in existing studies in the same area. Therefore, its planning scope and location are not present in the current reality, and we use it as a background to validate the method presented in this study. Therefore, it is unfortunate that satellite images could not be provided in this study to show the display of land use within the ecological corridor. But that doesn't mean our method isn't applicable within the planned corridors.
- Is there any regulatory framework to support this analysis?
- Response:
Thanks to the reviewer for your comments. Based on the existing research gaps in the construction of ecological corridors in metropolitan areas through quantitative adjustment of land use, this study proposed a new attempt, that is, an effective evaluation mechanism is established to measure the ecological, social, and economic comprehensive benefits of ecological corridors by adjusting existing land use in urban built environments. By maximizing the overall benefit index of the ecological corridor, we found the optimal ratio of agricultural land and constructed land located within the ecological corridor planning area to be converted to ecological land, which can guide the structural optimization of mixed land use and landscape within the ecological corridor planning area of the metropolitan area. We have verified the method in a typical metropolitan area in China and obtained reasonable and credible results. Although the analysis and results obtained in this study are not supported by the existing regulatory framework, they are an attempt of a new method, and the results can be connected with relevant local planning, to provide a quantitative and specific ecological corridor construction reference for practice. Relevant plans that can be connected include: "Suzhou Urban Green Space System Planning (2017-2035)", "Urban Master Plan of Jiaxing (2003-2020) (2017 Revision), and "Urban Master Plan of Huzhou (2017-2035)".
- The land use description is rather general: What types of built and natural environments are we dealing with? What is the relevance/impact of transportation infrastructures in the indexes provided?
- Response:
The reviewer was greatly appreciated for your comments and suggestions. With your reminder, we realized that our manuscript was too concise and general about the land use in the study area. Therefore, in the section "2.3.2 Other data sources", we have supplemented the description of land use in the study area, so that the built and natural environment of the study area can be more concretely understood by the reader. We have mainly supplemented the following aspects: 1) the subdivision and interpretation of three types of land use have been added; 2) supplemented with data on the proportion of three types of land in the study area; 3) The spatial pattern characteristics of the constructed land in the study area were described. The specific modifications are as follows:
“The agricultural land in the study area is mainly divided into paddy fields and dry land, the former refers to the agricultural land with guaranteed water sources and irrigation facilities, and is used to grow aquatic crops such as rice and lotus root. The latter refers to agricultural land that grows crops on natural precipitation and is mainly used to grow vegetables. They are a common type of agricultural land in the plain river network area, and also the type with the largest proportion of the total area of the study area (40.1%) of the three land use types. The ecological land in the study area can be subdivided into the following six categories: forest, grassland, river, lake and pond, wetland, and unused land, accounting for 35.5% of the total area of the study area. Constructed land includes urban land, rural residential land, transportation land, industrial land, and other construction land, accounting for 24.4% of the total area of the study area. The constructed land in the study area is distributed in the vast agricultural hinterland in a small concentrated and large scattered pattern.”
Although the subdivision of the three types of land use was introduced, in the results section, the "optimal conversion ratio" we present was still for the broad category of agricultural land or constructed land. That is, we didn't get the optimal conversion ratio for each land-use subdivision, such as the individual impact of "transportation infrastructures" on the index mentioned by the reviewer. On the one hand, this requires more refined indicators and detailed data to distinguish the differences between different subdivision of land use and the comprehensive benefits of ecological corridor construction, which may be difficult for us to obtain. On the other hand, this will undoubtedly make the overall model more complex and cumbersome, and considering that different countries and regions have different standards for land use subdivision, it may also bring limited model generalization. However, based on the reviewer's comments, we believed it is necessary to clarify the above issues in the "Limitations" section. Therefore, we have added the following to the "4.2 Limitations" section:
“Moreover, the result we presented was "optimal conversion ratio" for agricultural land or construction land. That is, we didn't get the optimal conversion ratio for every land-use subdivision. To do this, we need to set up finer indexes and obtain more detailed data to distinguish the differences between different subdivisions of land use on the overall benefits of ecological corridor construction, which brings difficulties for us. Also, this will undoubtedly make the overall model more complex and cumbersome, and considering that different countries and regions have different standards for land use subdivision, it may also bring about limited model promotion. However, exploring the contribution or impact of different land use subdivisions on the comprehensive benefits of ecological corridor construction can become a further research direction in the future.”
- Eventually, a language check is strongly suggested. Typos to amend (for example, fig. 2 "lengend").
- Response:
Many thanks to the reviewer for your comments and suggestions. First, we have proofread and corrected the text for the errors (including spelling and grammar). Secondly, we have commissioned native English-speaking colleagues in our research group to review and polish our manuscript, thus improving the language level of our manuscript to a certain extent.

Reviewer 2 Report
Dear Authors,
The article titled "Optimizing the landscape structure of hybrid land use in ecological corridors based on a comprehensive benefit index in a metropolitan area" aims to create a new index called the "comprehensive ecological corridor benefit index."
This indicator aims to quantify the comprehensive benefits in terms of ecological, social and economic aspects achieved by adjusting existing land use and building ecological corridors in urban development.
The research topic of the article is interesting and very important due to the fact that it deals with ecological corridors, provide urban ecological security. Ecological corridors play a very important role in ensuring the balance between urbanized areas and green areas.
After reading the article, I have the following comments and suggestions for improving the article:
Abstract
I suggest improving it by making it more readable. I suggest improving the abstract according to the guidelines of the journal Forests. There is no information about the methods used and the results of the study are not presented.
Introduction
In my opinion, it should be expanded to include the following news: why was this study undertaken? What studies have been conducted so far, where?
Material and methods
This subsection is very well prepared, including a diagram of the research procedure, a description of the research area with a map, and a detailed description of the methods. I consider the selection of research methods adopted for the study to be correct.
Theoretical bockgrund
This subsection is missing. It could possibly be replaced in the subsection Introduction. I suggest introducing more information on:
- What role the ecological corridor plays in the city, broken down by function f ecological, social, and economic.
Result.
The results are presented and described in a good way, they are very interesting and important for ecological use in cities.
The figures do not have sources given.
The authors should compare their project and results with the results of similar studies conducted on this topic in other parts of Europe and the world.
Technical errors to be removed:
Correct the literature according to the journal's rules.
[150, 182] suggests standardizing the notation of FIGURE
there are different notations in the text, e.g..
[148] "please see Figure 1
[155] (see Figure 2)
Best regards,
Reviewer
Author Response
Response to the reviewers’ comments
Reviewer 2:
- Abstract
I suggest improving it by making it more readable. I suggest improving the abstract according to the guidelines of the journal Forests. There is no information about the methods used and the results of the study are not presented.
- Response:
Thank you very much to the reviewer for raising this deficiency for us. Based on your suggestions, and with reference to the abstract structure of articles published in the journal Forests, we have revised the abstract: 1) refined the methodological section; 2) presented two main findings; 3) simplified background introduction. The specific modifications are as follows:
“To provide reference for the internal landscape structure adjustment of the ecological corridor composed of hybrid land use in the rapidly urbanized areas, first, we constructed the "Comprehensive benefit index of ecological corridors I" by using the three indexes of "Ecological benefit enhancement potential", “Economic input cost" and “Social coordination cost". Second, with the goal of maximizing the comprehensive benefits of the three aspects of ecological corridor construction, we established a functional relationship between the converted agricultural land area A, constructed land area C, and the index I, to determine the optimal proportion of agricultural lands and constructed lands converted into ecological lands within the planning scope of the ecological corridors. The results show that: 1) According to the conversion ratio, the ecological corridors in the study area can be divided into three degrees of conversion rate: low, moderate, and high; 2) Among the 66 ecological corridors, the agricultural lands in 26 ecological corridors and the constructed lands in 35 ecological corridors need to be converted into ecological land at a high ratio to ensure the high comprehensive benefits of the corresponding corridors.”
- Introduction
In my opinion, it should be expanded to include the following news: why was this study undertaken? What studies have been conducted so far, where?
- Response:
Thank you to the reviewer for your comments. Based on your suggestions, we have made the following additions in the Introduction section: 1) We further explained the need for land use adjustments to urban ecological corridors. 2) Some existing studies were listed to illustrate the current status of researches on ecological corridor construction through land use adjustment within ecological corridors. 3) Summarized the shortcomings of the above existing studies and put forward the necessity of this study. The specific changes are as follows:
1) “In highly urbanized areas, ecological lands are widely occupied, and it is difficult to maintain normal ecological processes only through the protection of existing ecological lands [36]. Therefore, it is sometimes necessary to adjust parts of the existing development lands or convert these non-ecological lands back into ecological lands to achieve the goal of ecological corridor construction [37].”
2) “Some studies have carried out corridor construction through the adjustment of land use within ecological corridors. On the basis of reconstructing the ecological security pattern of the Su-Xi-Chang metropolitan area, some scholars have proposed specific plans to reduce the construction land in the ecological corridor [40]. Other scholars have identified the total area where non-ecological land needs to be converted into ecological land in the main urban area of Chongqing [41]. Li, et al. [42] proposed that a 200-meter-wide corridor in Nanjing is ecologically suitable and economically sound for land use adjustment.”
3) “However, there is still a gap in understanding how to effectively incorporate existing land uses into the adjustment process within the designated width of ecological corridors, to achieve a balanced and maximized comprehensive benefit in the context of multidimensional and complex ecological, economic, and social environments. This requires establishing effective mechanisms to evaluate the comprehensive impact of ecological, economic, and social benefits on the construction of urban ecological corridors through land use adjustment.”
- Theoretical background
This subsection is missing. It could possibly be replaced in the subsection Introduction. I suggest introducing more information on:
- What role the ecological corridor plays in the city, broken down by function of ecological, social, and economic.
- Response:
Thanks to the reviewer for your suggestions. Based on your comments, we have added the theoretical background of ecological corridors in the Introduction section, including the following aspects: 1) the definition of ecological corridor, 2) the functional manifestation of ecological corridors in metropolitan areas (listed in three aspects: ecological, social, and economic), 3) the value of constructing ecological corridors. The specific modifications are as follows:
“Ecological corridors are strip-like or linear ecological landscapes with a certain spatial range [14,15], which are heterogeneous compared with the surrounding landscape [16]. They act as bridges and connections between various ecological source areas, and are also crucial for achieving urban sustainable development [17]. Unlike the linear transportation infrastructure and ecological roads in the city, which also play a role in connecting between destinations and deliver goods and materials, ecological corridors also play the functions of natural habitat, green open space, human habitat isolation , and human settlement environment beautification and enhancement [16,18], which are an inherent part of the landscape ecological network [16] and one of the concrete embodiments of the ecological environment in the metropolitan areas. Consequently, how to reasonably construct ecological corridors has become an important approach to maintaining connectivity between isolated habitat patches and mitigating habitat fragmentation in metropolitan areas [19].”
- Result.
1) The figures do not have sources given.
- Response:
Thanks to the reviewer for the reminder. All figures in our results section were drawn by the authors in ArcGIS software based on spatial data and analysis tools. We've added the following description below each figure in the Results section: “(Source: Drawn by the authors using ArcGIS software)”. In addition, since Figure 3 and Table 2 are data from the authors' team's existing studies, we labelled their sources as follows: “(Source: Shen, J.; Wang, Y. An improved method for the identification and setting of ecological corridors in urbanized area. Urban Ecosystems 2022, doi:10.1007/s11252-022-01298-5.)” and “(Source: Shen, J.; Wang, Y. An improved method for the identification and setting of ecological corridors in urbanized area. Urban Ecosystems 2022, doi:10.1007/s11252-022-01298-5.)”
2) The authors should compare their project and results with the results of similar studies conducted on this topic in other parts of Europe and the world.
- Response:
We appreciate the reviewer's suggestions. We believed it is important to compare the results of this study in southern China with those of similar studies conducted in other parts of the world, as there may be interesting differences in land use adjustment, comprehensive interest trade-offs, and ecological corridor construction in different countries and regions, which may provide us with necessary references when implementing relevant policies and planning on the ground. Unfortunately, we were unable to accomplish this in this study. First, through literature review, we found that how to adjust the ecological corridor based on existing land use within the planning width to achieve a balance and maximize its comprehensive benefits in ecological, economic, and social multi-dimensional and complex environments, which is still a gap in the current research. This is why we need to carry out this study. Second, through a search of the Web of Science database, we found some existing studies to build ecological corridors in metropolitan areas by quantitatively adjusting land use. However, much of this research were also done by Chinese scholars (e.g., cited in the literature [40-42]). If reviewers can provide us with some similar studies conducted in Europe or elsewhere, we would be more than willing to cite and compare them in the next round of the revision, and thank you very much for your help! We believed that the reviewer's suggestions are important, we have added the following expectations for future research in "4.2 Limitations":
“Therefore, in future study, it is necessary to compare the results of this study in south-ern China with the results of applying this method in other parts of the world, to find interesting differences in land use adjustment strategies and comprehensive benefit trade-off dimensions in different countries and regions, and obtain necessary references for the implementation of local policies and measures related to ecological corridor construction from these differences.”
- Technical errors to be removed:
1) Correct the literature according to the journal's rules.
- Response:
Thanks to the reviewer for your comments. We further checked the citation format of the references in the manuscript, and arranged and modified them according to the requirements of MDPI.
2) [150, 182] suggests standardizing the notation of FIGURE
- Response:
Thank the reviewer for pointing out the above problem for us. We have modified the notations into "Figure X".
3) there are different notations in the text, e.g. [148] "please see Figure 1, [155] (see Figure 2)
- Response:
Thank you to the reviewer for pointing out the above shortcomings for us. We have unified notations in the text, including: "please see Figure 1", "(please see Figure 2)", "(Please see Figure 6-b)", "(please see Appendix D)", "(please see Appendix F)", and "(please see Appendix H)".

Round 2
Reviewer 1 Report
The Authors met the revision requirements and the manuscript is now fit for publication.
Congrats to the Authors!